# RefCritic: Training Long Chain-of-Thought Critic Models with Refinement Feedback

## Abstract

With the rapid advancement of Large Language Models (LLMs), developing effective critic modules for precise guidance has become crucial yet challenging. In this paper, we initially demonstrate that supervised fine-tuning for building critic modules (which is widely adopted in current solutions) fails to genuinely enhance models' critique abilities, producing superficial critiques with insufficient reflections and verifications. To unlock the unprecedented critique capabilities, we propose RefCritic, a long-chain-of-thought critic module based on reinforcement learning with dual rule-based rewards: (1) instance-level correctness of solution judgments and (2) refinement accuracies of the policy model based on critiques, aiming to generate high-quality evaluations with actionable feedback that effectively guides model refinement. We evaluate RefCritic on Qwen2.5-14B-Instruct and DeepSeek-R1-Distill-Qwen-14B across five benchmarks. On critique and refinement settings, RefCritic demonstrates consistent advantages across all benchmarks, e.g., 6.8% and 7.2% gains on AIME25 for the respective base models. Notably, under majority voting, policy models filtered by RefCritic show superior scaling with increased voting numbers. Moreover, despite training on solution-level supervision, RefCritic outperforms step-level supervised approaches on ProcessBench, a benchmark to identify erroneous steps in mathematical reasoning.

## 1 Introduction

In recent years, Large Language Models (LLMs) have demonstrated remarkable capabilities in executing complex reasoning tasks such as mathematical problem-solving and code generation Yang et al. (2025a); Hui et al. (2024). As these models continue to evolve, their reasoning processes have grown increasingly sophisticated, encompassing multiple elaborate steps and diverse pathways (Guo et al., 2025; Team, 2025). This progression introduces a critical challenge: reasoning processes are becoming substantially difficult for humans to supervise, making errors within these intricate chains harder to identify and rectify. The escalating complexity of LLM-generated solutions necessitates more effective analytical frameworks to evaluate and enhance reasoning quality, extending beyond the constraints of human supervisory oversight.

Developing LLM critics has become a promising direction for evaluating complex reasoning tasks (Liu et al., 2025; Zhang et al., 2024; Mahan et al., 2024; Wang et al., 2023; Ankner et al., 2024), functioning as specialized modules to analyze reasoning processes and identify errors. Ideally, LLM critics are expected to comprehensively analyze content generated by the policy model, delivering targeted critiques that identify logical inconsistencies or factual errors and improve the refinement quality of the policy model. However, contemporary approaches exhibit two critical limitations. Firstly, they frequently produce superficial evaluations characterized by insufficient analytical depth (Zheng et al., 2025; Tang et al., 2025a) and typically necessitate granular step-level annotations of the solution for optimization (Yang et al., 2025b). Secondly, current implementations mainly focus on metrics of critic performance while overlooking the practical utility of critiques in enhancing policy model refinement.

In this paper, we propose RefCritic, a long chain-of-thought critic model with refinement feedback to tackle the above limitations, which could generate in-depth critiques that not only achieve superior critic performance but also effectively guide policy model refinement through actionable feedback. This process begins with prompting open-source models (e.g., DeepSeek-R1-Distill-Qwen)

Figure 1: The Critic model with Refinement Feedback RefCritic framework consists of two steps: (1) cold-start via rejective sampling fine-tuning, (2) rule-based reinforcement learning with refinement feedback. With this two-stage optimization, RefCritic generates in-depth critiques that achieve superior critic performance and effectively guide policy model refinement through actionable feedback.

to generate seed data containing three essential components: long CoT analysis, solution validity judgments, and refinement suggestions. After rigorous quality filtering based on judgment accuracy, about 10K valid samples are obtained, which are utilized to establish cold-start critic models via Supervised Fine-Tuning (SFT). We then conduct preliminary assessments to verify whether SFT is sufficient for producing comprehensive critiques. Results reveal that SFT alone struggles to produce in-depth critiques despite generating lengthy CoT content, as models frequently exhibit misleading analytical patterns where correct judgments emerge from flawed reasoning processes (a persistent issue observed in existing LLM critics), resulting in unreliable performance evaluations. Furthermore, the absence of explicit policy model interaction during SFT leads to critiques that inadequately prioritize feasible guidance for effective refinement. The above observations highlight the difficulty of SFT in producing critiques with both accurate evaluations and practical feedback for refinement.

To further enhance the critic reliability and establish a causal connection between critique quality and policy refinement outcomes, we implement a dual-reward reinforcement learning framework based on cold-start models for finalizing RefCritic. The first reward signal stems from the instance-level binary accuracy metric (0/1 values) for evaluating the critic models in solution judgment capability. The second reward quantifies policy model improvement through accuracy gains after incorporating refinement suggestions. Critically, the second reward establishes an explicit feedback loop where the quality of effective critique is operationally defined by its capacity to drive measurable enhancements to the policy model. This dual-reward design ensures that high-reward critiques are those that not only accurately identify solution flaws but also provide actionable guidance leading to verifiable performance gains.

We validate the effectiveness of RefCritic on multiple challenging mathematical datasets: AIME24, AIME25, and Olympiad (He et al., 2024). In the refinement after critique setting, feedback generated by RefCritic based on Qwen2.5-14B and DeepSeek-R1-Distill-Qwen-14B consistently enhances the corresponding policy models' performance, with improvements of 6.8% and 7.2% on AIME25, and 9.9% and 2.6% on Olympiad, respectively. In the majority vote with critique setting, RefCritic demonstrates increasingly significant performance gains as the sampling count increases. With 64 samples, RefCritic achieves an average improvement of 3.6 percentage points on AIME25 compared to scenarios without critique, consistently outperforming other critique baselines. Moreover, RefCritic effectively enhances majority vote performance even when applied to more powerful models, like QwQ and DeepSeek-Distill-Qwen-32B. These performance improvements indicate that our dual reward mechanism successfully aligns critique generation with both evaluation accuracy and

refinement utility, enabling critic models to produce not only precise solution assessments but also actionable feedback that effectively guides refinement processes. Furthermore, it is worth noting that RefCritic generalizes effectively to step-level critique tasks without requiring any step-level labels during training, achieving remarkable performance on ProcessBench (Zheng et al., 2024).

## 2 RELATED WORK

**Test-time Scaling** Test-time scaling techniques have emerged as a powerful approach to enhance LLM reasoning capabilities through increased computational resources during inference (Charniak & Johnson, 2005; Snell et al., 2024; Wu et al., 2024; Yao et al., 2023; Chen et al., 2024; Jaech et al., 2024; Guo et al., 2025; Team, 2025). The effectiveness of these approaches can be improved by judgment or verification mechanisms. Besides traditional process reward models (PRMs) that directly predict numerical correctness scores for solution steps (Uesato et al., 2022; Lightman et al., 2023; Zheng et al., 2025; 2023; Chen et al., 2025; Zhang et al., 2025), recent methods frame judgement as language generation tasks that offer greater interpretability and scalability (Liu et al., 2025; Zhang et al., 2024; Mahan et al., 2024; Wang et al., 2023; Ankner et al., 2024). Among them, a promising approach is LLM Critics, which uses LLMs as critic models to verify solutions (McAleese et al., 2024; Zheng et al., 2025; Yang et al., 2025b).

**Critic Model** The judgment ability of LLMs has garnered significant research interest due to their potential to enhance reasoning through explicit error detection and correction guidance (Lan et al., 2024; Lin et al., 2024; Zheng et al., 2024). Current approaches fall into two main categories: LLM-as-a-Critic (Zheng et al., 2025; Yang et al., 2025b) leverages off-the-shelf models through careful instruction design, and specialized critic models (McAleese et al., 2024; Lan et al., 2024; Shi & Jin, 2025; Khalifa et al., 2025; Yang et al., 2025b) that employ fine-tuning or reinforcement learning to enhance judgment ability. While recent research emphasizes that critique effectiveness should be validated through correction outcomes (Tang et al., 2025b), most existing critic models focus exclusively on critique and ignore the future benefit it can bring to refinement. Some concurrent works, including RCO (Yu et al., 2025), CTRL (Xie et al., 2025), and MultiCritique (Lan et al., 2025), have started to explore this direction. RCO proposes a refinement-oriented training paradigm that relies on preference judgments that may introduce alignment biases in tasks requiring precise verification. Similarly, CTRL (Xie et al., 2025) utilizes reinforcement learning to optimize critics for code generation, prioritizing refinement success rates over explicit judgment accuracy. MultiCritique (Lan et al., 2025) leverages refinement outcomes to filter multi-agent feedback for constructing static critique datasets. In contrast, RefCritic employs a dual-reward mechanism with ground-truth-based verification, specifically optimized for mathematical reasoning tasks where objective correctness is paramount.

## 3 SFT IS INSUFFICIENT FOR DEEP CRITIQUES

To better understand the challenges in developing effective critic models, we first examined a straightforward approach widely adopted in previous research: supervised fine-tuning with rejection sampling. This approach has demonstrated success in improving judgment capabilities of critics in several studies (Tang et al., 2025a; Zheng et al., 2025; Khalifa et al., 2025).

Specifically, we employed Qwen2.5-14B-Instruct/DeepSeek-Distill-Qwen-14B as our policy model and DeepSeek-Distill-Qwen-32B as the critic model[1]. Employing rejection sampling, we collected approximately 10K critique training samples from a subset of NuminaMath. Each training sample comprised a problem statement, model response, chain-of-thought reasoning, judgment on solution correctness, and refinement suggestions. We subsequently fine-tuned the policy models on these datasets to develop critique capabilities. To evaluate the effectiveness of these fine-tuned critic models, we tested them on responses collected from AIME25, assessing their ability to identify errors and provide feedback that could meaningfully improve policy model performance.

Our experiments revealed a significant disparity between the critic model capabilities and their practical utility. As shown in Table 1, whether it is a Qwen-based critic or DeepSeek-Distill-Qwen-based critic, SFT-trained models significantly outperformed self-critique approaches in critique accuracy

---

[1]For Qwen2.5-14B-Instruct, we provide an empty thinking process and only use the content after "<think>".

metrics. However, when these critiques were used to refine policy model outputs, the resulting performance improvements were minimal, sometimes even inferior to those achieved through self-critique methods. This counterintuitive result suggests that conventional evaluation metrics for critics may not align with their actual utility in improving model performance.

Further analysis of the SFT model outputs revealed two critical limitations. First, critics often arrive at correct judgments through flawed or superficial reasoning processes, creating a false impression of reliability despite inconsistent analytical quality. This problem is particularly evident in Qwen-based models, as the critique length after SFT showed no significant increase, with an average length of less than 500 tokens. Second, the feedback typically identified error locations but lacked specific, actionable guidance for improvement. Critics frequently offered vague suggestions or restated problem requirements rather than providing concrete directions for correcting mathematical misconceptions or reasoning flaws. These findings directly support our hypothesis that conventional SFT approaches, while successful in training critics to make binary judgments, fail to develop models that can provide the actionable, improvement-oriented feedback necessary for effective solution refinement.

| Method | Critique Accuracy | Pass@1 after Refinement |
|---|---|---|
| *Qwen2.5-14B as Base Model* | | |
| Pass@1 | - | 14.4 |
| Self-Critique | 51.8 | 14.5 |
| SFT | 80.6 | 15.0 |
| *R1-Qwen-14B as Base Model* | | |
| Pass@1 | - | 49.2 |
| Self-Critique | 71.5 | 52.1 |
| SFT | 78.9 | 51.4 |

Table 1: Preliminary experiment on AIME25 to verify whether SFT can emerge deep critic. We can see that although the SFT model achieves strong performance in critique evaluation, incorporating its feedback into the policy model yields only marginal performance gains. R1-Qwen represents DeepSeek-Distill-Qwen.

## 4 REFCRITIC

We propose RefCritic, a novel approach for developing effective critic modules that provide actionable feedback for mathematical reasoning tasks. As illustrated in Figure 1, RefCritic employs a two-stage methodology. First, we develop a cold-start critic model via supervised fine-tuning that activates the model's reasoning judgment capabilities and enables structured output generation. Second, we introduce a rule-based reinforcement learning framework with dual rewards optimizing critics for both solution-level correctness and refinement effectiveness, measured by concrete improvements in policy model performance. This dual-reward mechanism ensures our critic models not only accurately evaluate solutions but also provide guidance that leads to substantive improvements in reasoning capabilities.

When faced with complex tasks such as critique generation, LLMs often exhibit problematic behaviors, including instruction unfollowing (He et al., 2025) and answer leakage (Yang et al., 2025b). To address these challenges, we implement SFT with rejection sampling to standardize model outputs. Following our preliminary experimental setup, we leverage a more powerful model to generate initial critic responses, then systematically filter out responses containing erroneous judgments, instruction violations, or solution leakage risks through rule-based screening. This curated dataset serves as the foundation for our LLM training, ensuring the resulting critic model adheres to desired output formats while maintaining evaluation integrity.

Despite the effectiveness of the supervised fine-tuning approach in producing format-compliant critics, the preliminary experiments revealed fundamental shortcomings that limit practical utility. As previously demonstrated, SFT models exhibit a misleading combination of accurate judgments built upon superficial reasoning, alongside feedback that identifies errors without providing actionable remediation strategies. Rather than reiterating these limitations, we recognize them as symptomatic of a deeper issue: conventional training methods optimize for solution classification accuracy rather than refinement capability. This insight motivates us to shift from purely supervised learning toward a reinforcement learning framework that explicitly rewards critics not just for evaluation correctness, but for generating feedback that demonstrably improves subsequent solutions.

To address these limitations, we introduce a dual-reward reinforcement learning framework that optimizes both judgment accuracy and refinement effectiveness. Our approach evaluates critics

based on two key metrics: (1) their ability to correctly classify solutions as right or wrong, and (2) the improvement when the policy model uses their feedback to refine incorrect solutions.

This framework ensures our critics develop both strong evaluation capabilities and the ability to generate constructive feedback that leads to measurable improvements in reasoning outcomes.

RefCritic uses GRPO (Group Relative Policy Optimization) to train the critic model. Let $\pi_c$ denote the critic model, and $\pi_\theta$ denote the policy model, used for generating critiques and solutions separately. For a question $x$ from dataset $\mathcal{D}$ with golden answer $a$, the policy model generates an initial solution $y \sim \pi_\theta(y|x)$. The correctness of this solution is determined by $c = \mathbb{I}[y = a] \in \{0(\text{incorrect}), 1(\text{correct})\}$, where $\mathbb{I}[\cdot]$ is a rule-based discriminator that determines whether the generated answer matches the golden answer.

Then the critic model generates $G$ critiques $\pi_c(x, y) \to s = (z, \hat{c})$, where $z$ represents the extensive reasoning process and $\hat{c} \in \{0, 1\}$ represents $\pi_c$'s judgment of response correctness.

For any critique $s_i$, its critique reward $r_i^c$ is assigned as:

$$r_i = \begin{cases} 1 & \text{if } \hat{c} = c \\ 0 & \text{otherwise} \end{cases} \tag{1}$$

We apply the following objective to train the policy model:

$$J = \sum_{i=1}^{G} \frac{1}{G|s_i|} \sum_{t=1}^{|s_i|} \left\{ \min\left[ ratio_i \hat{A}_{i,t}, \ \text{clip}\left(ratio_i, 1 - \epsilon, 1 + \epsilon\right) \hat{A}_{i,t} \right] - \beta D_{KL} \right\} \tag{2}$$

$$ratio_i = \frac{\pi_\theta(s_{i,t}|y, s_{i,<t})}{\pi_{\theta_{\text{old}}}(s_{i,t}|y, s_{i,<t})} \tag{3}$$

where $G$ represents the number of critiques generated by $\pi_c$ for each original response $y$, $s_i$ represents the i-th generated critique, and $\hat{A}_{i,t}$ represents the token-level advantage. GRPO eliminates the need for a value function through group relative advantage estimation:

$$\hat{A}_i = \frac{r_i^c - \text{mean}(\{r_1^c, r_2^c, \ldots, r_G^c\})}{\text{std}(\{r_1^c, r_2^c, \ldots, r_G^c\})} \tag{4}$$

where $r_i^c$ represents the critique reward of the i-th critique. When optimizing $\pi_c$, $r_i^c$ is determined by whether $s_i$ contains a correct judgment $\hat{c}_i$.

However, this design has issues because the $\pi_c$ may not provide useful feedback for $\pi_\theta$ to refine its solution. Additionally, $\pi_c$ might arrive at correct judgments through incorrect reasoning (this is because we don't use step-level supervision signals like PRM, only solution-level signals). Therefore, we introduce refinement feedback $r^r$ to enhance $\pi_c$'s Reward. To obtain $r^r$, we use $\pi_\theta$ to generate $N$ refined solutions $\{\hat{y}_i\}_{i=1}^{N}$ based on each critique from $\pi_c$, where $\hat{y} = \pi_\theta(x, y, s)$. Then, we have:

$$r_r = \frac{1}{m} \sum_{i=1}^{m} \mathbb{I}[\hat{y}_i = a] \tag{5}$$

The imporved reward can be calculated as:

$$R = \frac{r^c + \lambda r^r}{1 + \lambda} \tag{6}$$

where $\lambda$ is a hyperparameter that balances the importance between judgment accuracy and refinement effectiveness, a higher $\lambda$ value places greater emphasis on the critic's ability to provide actionable feedback that leads to correct solutions, while a lower value prioritizes accurate solution classification. To prevent reward imbalance during reinforcement learning, we divide the reward by $1 + \lambda$.

Notably, RefCritic only calculates Refinement Reward for solutions that satisfy two conditions: (1) the solution is incorrect, and (2) the critic correctly identifies it as incorrect. This selective approach

is justified because refinement is unnecessary in these scenarios: when the solution is already correct, or when the critic fails to detect an incorrect solution. Besides, as explained in Appendix E, this design will significantly reduce the cost of applying the refinement reward. Therefore, we have the final reward can be calculated as:

$$R = \begin{cases} \frac{r^c + \lambda r^r}{1 + \lambda} & \text{if } c = 0 \text{ and } \hat{c} = 0 \\ r^c & \text{otherwise} \end{cases} \tag{7}$$

where $R_c = r$ represents whether the critique $s$ contains a correct judgment $\hat{c}$, ensuring that $\pi_c$ can accurately identify solution correctness. $R_r$ represents whether $\pi_c$'s critique provides useful feedback for $\pi_\theta$. The final advantage estimation is:

$$\tilde{A}_i = \frac{R_i - \text{mean}(\{R_1, R_2, \ldots, R_G\})}{\text{std}(\{R_1, R_2, \ldots, R_G\})} \tag{8}$$

In summary, our RefCritic framework alleviates the key limitations of existing critic models through this dual-reward reinforcement learning approach. By explicitly optimizing for both judgment accuracy and refinement effectiveness, we develop critics that not only accurately evaluate mathematical solutions but also provide actionable feedback that leads to concrete improvements in reasoning outcomes.

## 5 EXPERIMENTS

### 5.1 EXPERIMENTAL SETUP

**Models**  For our implementation of RefCritic, we utilize Qwen2.5-14B-Instruct (Yang et al., 2025a) and DeepSeek-R1-Distill-Qwen-14B (Guo et al., 2025) as the backbone. In our framework, these models perform two distinct functions: first, as policy models that generate solutions for mathematical problems; and second, as the foundation models to develop our critic models.

**Data Construction**  We construct our training dataset by filtering approximately 120k high-quality mathematical problems from the 900k problems in NuminaMath-1.5 (LI et al., 2024). Our filtering pipeline includes: (1) deduplication through exact string matching and semantic similarity; (2) problem filtering; (3) difficulty balancing. For critic training, we sample 8 responses per problem and retain at most one correct and one incorrect per problem to ensure balanced training data. Detailed process can be found in Appendix C.

**Benchmarks**  We evaluate the performance of RefCritic on challenging mathematical benchmarks, including AIME 2024/2025 (American Invitational Mathematics Examination problems), and OlympiadBench (He et al., 2024). Since RefCritic was trained only on math problems, we conduct out-of-distribution tests on the code generation task LiveCodeBench (Jain et al., 2024) and the science QA benchmark GPQA-Diamond (Rein et al., 2024). Furthermore, to evaluate RefCritic's capability for fine-grained error localization, we leverage ProcessBench (Zheng et al., 2024) to assess its ability to accurately identify the specific step where an error occurs.

**Training Details**  In the SFT stage, we train the critic models with a learning rate of 7e-6 and a batch size of 512 for three epochs. For the RL stage, we employ the GRPO algorithm (Shao et al., 2024) to enhance critic performance. We sample 8 critics for each input, each rollout comprising 128 inputs, and conduct on-policy training with a learning rate of 1e-6. We set the maximum sequence length to 8K and 16K tokens for Qwen2.5-Instruct and DeepSeek-R1-Distill-Qwen, respectively. For refinement feedback, we use policy models to generate 8 refinements for each critic. Considering the cost of sampling refinements, we initially set $\lambda=0$ to achieve rapid improvement in critic performance for 600 steps, where no refinement is generated. We subsequently adjust to $\lambda=1$ to balance the trade-off between the two reward components and continue training for 300 steps.

**Evaluation Details**  For evaluation, both policy and critic models apply temperature=0.6 and $top\_p$=0.95. For AIME24/25, we pre-sample 128 responses as the response pool for performance calculations. For OlympiadBench/GPQA/LiveCodeBench, we sample only 32 responses due to their larger scale. Then, we randomly select responses from the response pool for metric calculation and report the average results over 1000 trials. We adopt multiple evaluation settings.

| Model | AIME25 | | | AIME24 | | | Olympiad | | |
|---|---|---|---|---|---|---|---|---|---|
| | $Pass_r$@1 | $Maj_c$@8 | $Maj_c$@64 | $Pass_r$@1 | $Maj_c$@8 | $Maj_c$@64 | $Pass_r$@1 | $Maj_c$@8 | $Maj_c$@16 |
| *Qwen-14B as policy model* | | | | | | | | | |
| Qwen-14B Majority Vote *(No Critic)* | 14.4 | 19.2 | 23.3 | 13.7 | 16.5 | 16.6 | 45.8 | 52.2 | 53.6 |
| Qwen-14B Self-Critic | 14.5 | 19.1 | 22.7 | 13.7 | 18.5 | 21.2 | 45.8 | 52.3 | 54.0 |
| DeepCritic (Yang et al., 2025b) | 13.0 | 17.4 | 19.4 | 13.4 | 18.8 | 23.7 | 43.6 | 51.7 | 52.7 |
| ThinkPRM-14B (Khalifa et al., 2025) | 16.1 | 20.1 | 22.3 | 16.8 | 20.7 | **26.9** | 49.7 | 54.7 | 56.6 |
| RefCritic-Qwen-14B(SFT) | 15.0 | 19.3 | 23.4 | 15.2 | 19.2 | 23.9 | 46.6 | 52.5 | 54.3 |
| RefCritic-Qwen-14B(RL) | **21.2** | **21.5** | **24.4** | **23.0** | **21.4** | 26.6 | **55.7** | **57.3** | **59.2** |
| *R1-Qwen-14B as policy model* | | | | | | | | | |
| R1-Qwen-14B Majority Vote *(No Critic)* | 49.1 | 61.6 | 62.0 | 67.6 | 78.7 | 80.1 | 77.7 | 82.7 | 83.3 |
| R1-Qwen-14B Self-Critic | 50.0 | 60.6 | 62.9 | 70.5 | 79.3 | 82.4 | 78.8 | 82.7 | 83.3 |
| DeepCritic (Yang et al., 2025b) | 49.1 | 58.0 | 59.0 | 67.3 | 77.2 | 78.8 | 76.1 | 81.6 | 82.1 |
| ThinkPRM-14B (Khalifa et al., 2025) | 43.9 | 58.7 | 61.2 | 62.6 | 76.3 | 81.0 | 75.2 | 82.1 | 82.7 |
| RefCritic-R1-14B(SFT) | 51.3 | 61.6 | 62.8 | 71.4 | 79.4 | **83.1** | 78.7 | 83.0 | 84.4 |
| RefCritic-R1-14B(RL) | **56.3** | **65.2** | **68.1** | **72.8** | **80.4** | 82.5 | **80.3** | **83.9** | **84.7** |

Table 2: Performance comparison of different approaches on AIME24/25 and Olympiad. **Qwen-14B Majority Vote** and **R1-Qwen-14B Majority Vote** denote the baseline that performs majority vote directly. $Pass_r$ indicates the performance after one round of critique and refinement. $Maj_c$ indicates the majority vote performance after using the critic filtering solutions. Considering the cost of sampling refinements, we train refcritic with two RL training stages: first, train without refinement reward to achieve rapid improvement in critic performance; then, train with refinement reward to optimize the ability to generate valuable feedback for refinement.

- *Majority Vote with Critique($Maj_c$@N)*: The critic first evaluates the $N$ sampled solutions and filters out those judged as incorrect. We then apply a majority vote on these remaining solutions to find the final answer.

- *Refinement after Critique($Pass_r$@1)*: The policy model generates an initial solution, which is then critiqued by the critic. If judged as incorrect, the policy model refines the solution based on the feedback. We report the pass@1 accuracy of the final refined answer.

- *Process Critique Evaluation*: For process-level evaluation, since our critic models were trained to output natural language critiques rather than explicit step indices, we use Qwen2.5-14B-Instruct to identify the step index the critic judges incorrect.[2] Following ProcessBench, we report the F1 score, which is the harmonic mean of precision for correct and incorrect solutions.

## 5.2 MAIN RESULTS

As shown in Table 2, we present the performance of RefCritic against various baselines on AIME24, AIME25, and OlympiadBench datasets. In the one-round critique and refinement settings, RefCritic consistently provides the most effective feedback for policy model improvement, demonstrating the effectiveness of incorporating refinement performance as a reward in our reinforcement learning approach. Specifically, on the challenging AIME25 dataset, RefCritic-Qwen-14B and RefCritic-R1-14B enhance the policy model's $Pass$@1 performance by 6.8% and 7.2%, respectively, significantly outperforming both self-critique baselines and models trained via supervised fine-tuning. Similar patterns emerge across AIME24 and Olympiad benchmarks, confirming that directly optimizing for policy model refinement performance during RL training enables critic models to generate more actionable feedback.

When scaling up the policy model's response generation and applying critic model filtering, RefCritic achieves superior performance across nearly all experimental settings. For instance, on AIME25, RefCritic-RL improves $Maj_c$@64 with an average benefit of 3.6%(1.1% for RefCritic-Qwen and 6.1% for RefCritic-R1). These results demonstrate that our refinement-oriented critic not only enhances feedback quality but also improves critical evaluation capabilities. Notably, as the sampling scale increases from 8 to 64, the overall performance gains from RefCritic become more pronounced, indicating our critic models' high discriminative accuracy in identifying and preserving high-quality solutions from larger candidate pools. We also propose some ablation studies to understand the role of the two RL training stages in Appendix B and Appendix D.

---

[2]We only provide the solution and critique, without the corresponding problem. Every critic we evaluated would go through this process.

| Model | GSM8K | MATH | Omni-Math | Olympiad | Avg. |
|---|---|---|---|---|---|
| *PRM* | | | | | |
| Math-Shepherd-PRM-7B* | 47.9 | 29.5 | 24.8 | 23.8 | 31.5 |
| RLHFlow-PRM-8B-DS* | 38.8 | 33.8 | 16.9 | 16.9 | 26.6 |
| Qwen2.5-Math-PRM-7B* | 68.2 | 62.6 | 50.7 | 44.3 | 56.5 |
| *Prompt LLM as Critic* | | | | | |
| Qwen2.5-14B-Instruct | 61.7 | 52.6 | 41.3 | 43.1 | 49.7 |
| Qwen2.5-72B-Instruct | 74.6 | 61.8 | 51.7 | 52.8 | 60.2 |
| R1-Qwen-7B | 75.3 | 74.4 | 56.9 | 63.5 | 67.5 |
| R1-Qwen-14B | 75.9 | 76.2 | 59.6 | 63.6 | 68.8 |
| GPT-4o-0806* | 79.2 | 63.6 | 51.4 | 53.5 | 61.9 |
| *Baseline Critic* | | | | | |
| SCRIT-72B (Tang et al., 2025a) | 80.2 | 60.0 | 32.5 | 27.8 | 50.0 |
| DeepCritic-7B (Yang et al., 2025b) | 72.6 | 72.8 | 56.0 | 60.9 | 65.6 |
| ThinkPRM-14B (Khalifa et al., 2025) | 67.6 | 71.4 | 54.8 | 59.3 | 63.3 |
| *Our Critic* | | | | | |
| RefCritic-Qwen-14B | 81.9 | 71.2 | 58.1 | 60.7 | 68.0 |
| RefCritic-R1-14B | **86.3** | **82.0** | **67.6** | **72.3** | **77.1** |

Table 4: The evaluation results of PRMs, LLM as a critic, and RefCritic critic models on ProcessBench. The metric is the F1 score, the harmonic mean of precision for correct and incorrect solutions. All our critic models are followed by an extract model (Qwen2.5-14B-Instruct) to get the error step for easy evaluation. Content marked with "*" sourced from Processbench. As shown in Appendix G, we use the same template as used in Processbench.

## 5.3 OUT-OF-DISTRIBUTION PERFORMANCE

Additionally, we evaluated RefCritic on out-of-distribution tasks. Considering that the model was trained on mathematical data, we chose to use Live-CodeBench to verify its performance on coding, and GPQA to evaluate its performance in challenging knowledge reasoning. We found that RefCritic still performs well on out-of-distribution benchmarks. Although the improvements are not as substantial as in the mathematical tasks, they still bring considerable gains. Specifically, RefCritic-R1-14B achieved a 3.1% performance improvement on LCB[3], and improved $Maj_c@64$ from 61.6% to 65.1% on the GPQA task, representing a 3.5% performance gain. Similar progress also appeared in RefCritic-Qwen-14B. These results suggest that RefCritic's critic capabilities can be applied to a wide range of tasks.

| Model | LCBench | GPQA | |
|---|---|---|---|
| | $Pass_r@1$ | $Pass_r@1$ | $Maj_c@16$ |
| *Qwen-14B as policy model* | | | |
| Qwen-14B Majority Vote | 18.9 | 19.5 | 23.3 |
| Qwen-14B Self-Critic | 20.9 | 19.5 | 22.7 |
| RefCritic-Qwen-14B(SFT) | 21.5 | 19.2 | 22.8 |
| RefCritic-Qwen-14B(RL) | **22.9** | **20.0** | **24.3** |
| *R1-Qwen-14B as policy model* | | | |
| R1-Qwen-14B Maj | 51.0 | 58.7 | 61.6 |
| R1-Qwen-14B as Critic | 52.4 | 57.7 | 60.6 |
| RefCritic-R1-14B(SFT) | 52.3 | 58.0 | 62.5 |
| RefCritic-R1-14B(RL) | **54.1** | **59.3** | **65.1** |

Table 3: Performance comparison of different approaches on LiveCodeBench and GPQA.

## 5.4 CRITIC PERFORMANCE

In this section, we evaluate RefCritic on ProcessBench to explore whether it can accurately identify true error locations in solutions. The experimental results presented in Table 4 demonstrate that RefCritic significantly outperforms most previous baselines, including methods that utilize step-level supervision. RefCritic-Qwen achieves an average performance of 68, while RefCritic-R1 reaches an impressive 77 average performance. This indicates that our dual reward mechanism effectively guides the model in developing accurate error identification capabilities.

This finding is consistent with the growth of the model output length during RL training. Specifically, the average output length of RefCritic-Qwen increased from about 500 tokens to 3500 tokens, while RefCritic-R1 increased from 3000 tokens to 8000 tokens. This indicates the increasingly detailed critiques, making step-level critique possible.

These findings demonstrate that even without explicit step-level supervision, our approach enables critic models to develop a nuanced understanding of solution processes and identify errors with high

---

[3]Since coding tasks cannot perform Majority Vote, we only report $Pass_r@1$ performance.

precision. This capability is crucial for generating actionable feedback that can effectively guide policy models toward improved solutions.

## 5.5 TEST-TIME SCALING

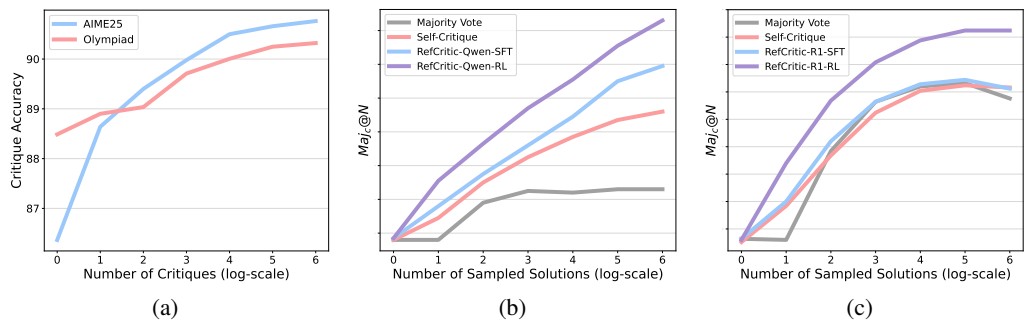

Figure 2: Test-time scaling with RefCritic. **Subplot (a)**, scaling the number of critiques with RefCritic-R1-14B, DeepSeek-R1-Distill-Qwen-14B as policy model. **Subplot (b)**, scaling the number of sampled solutions with RefCritic-Qwen-14B on AIME24, measured by $Maj_c@N$(majority vote after critique), Qwen2.5-14B-Instruct as policy model. **Subplot(c)**, scaling the number of sampled solutions with RefCritic-R1-14B on AIME25, measured by $Maj_c@N$, DeepSeek-R1-Distill-Qwen-14B as policy model.

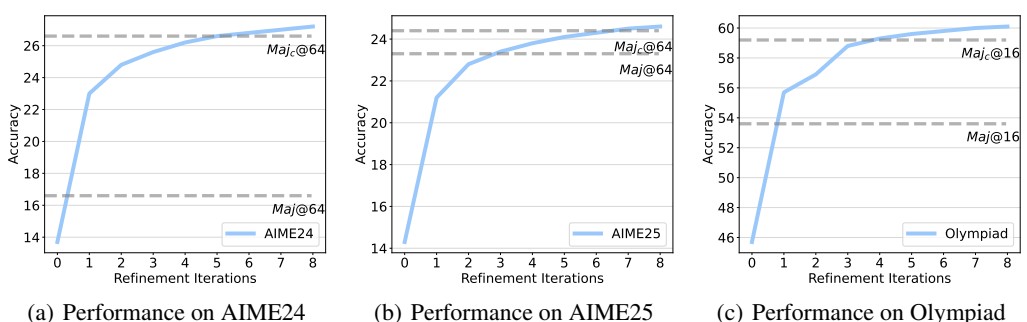

(a) Performance on AIME24    (b) Performance on AIME25    (c) Performance on Olympiad

Figure 3: Test-time scaling of iterative refinement with RefCritic-Qwen-14B. We iteratively perform critique and refinement until RefCritic judges the policy model's solution to be correct, or reaches the maximum iterations (8).

We investigate whether RefCritic's critique capabilities and model performance benefit from test-time scaling. We construct evaluation sets by sampling 16 solutions per problem from the AIME25 and Olympiad datasets. For each question, we sample 64 critiques from RefCritic to examine whether increasing critique quantity progressively enhances critique performance. Figure 2 (a) shows that RefCritic's critique performance improves consistently with increased sampling accounts. On AIME25, performance increases steadily, achieving a 4% improvement with more sampled critiques. This suggests that RefCritic benefits from test-time scaling through critique aggregation, thereby enhancing its accuracy in critiquing mathematical solutions. The Olympiad scaling curve is relatively flat, likely due to task difficulty for the policy model, though a clear scaling trend remains visible. Figure 2 (b) and (c) illustrate the scaling performance of RefCritic-Qwen-14B and RefCritic-R1-14B on AIME25, measured by $Maj_c@N$ across increasing rollout samples. The results demonstrate that RefCritic consistently outperforms baselines across different sampling scales, with performance gaps broadening as sample numbers increase. This validates that our refinement-oriented critic approach maintains effectiveness advantages in large-scale inference scenarios.

Beyond test-time scaling via solution sampling, we further investigate the potential of RefCritic in an iterative refinement setting. For each solution generated by the policy model, we establish a loop of critique and refinement. This process continues iteratively until RefCritic judges the solution to be correct or the maximum iteration limit (8) is reached. We evaluate the RefCritic-Qwen-14B model on AIME24, AIME25, and Olympiad. Consistent with the main experiment, we sample 128

initial solutions per problem for AIME and 32 for Olympiad. We report the average accuracy at each round. Figure 3 presents the detailed performance trajectory across iterations. The results demonstrate that iterative refinement yields significant performance gains. For instance, on AIME24, the pass rate doubles from a baseline of 13.6% at Round 0 to 27.2% at Round 8. Similarly, AIME25 and Olympiad exhibit substantial improvements, climbing to 24.5% and 60.1%, respectively. Remarkably, with only 8 rounds of iterative refinement, RefCritic achieves performance comparable to or even surpassing the $Maj_c@64$ baseline (majority vote with critic). This indicates that guiding the model to self-correct is a highly efficient alternative to massive sampling. However, we observe that the performance improvements begin to saturate between rounds 6 and 8. Through manual inspection of the failure cases in later rounds, we find that this is because: 1) after several iterations, most problems solvable by the Policy Model have been addressed, and the remaining problems are beyond the model's scope, and 2) RefCritic tends to misjudge the correctness of refined solutions that follow its own feedback, which is an interesting problem we may look into in the future.

## 5.6 SUPERVISION OF STRONGER MODELS

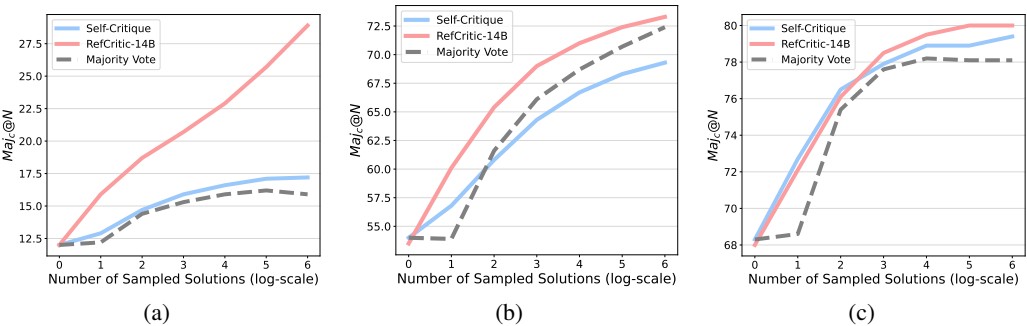

(a)  (b)  (c)

Figure 4: Supervision of RefCritic-R1-14B on stronger models. **Subplot (a)**, Qwen2.5-72B-Instruct as policy model. **Subplot (b)**, DeepSeek-Distill-Qwen-32B as policy model. **Subplot (c)**, QwQ-32B as policy model.

We also investigate whether RefCritic can provide effective cross-model supervision for more powerful reasoning models. We evaluate our approach using stronger models (QwQ, DeepSeek-Distill-Qwen-32B, and Qwen 2.5-72B) on the AIME25 dataset. We compare three settings: (1) standard majority voting, (2) self-critique where models evaluate their own solutions, and (3) cross-model supervision using RefCritic-14B as critic. Figure 4 shows that even the most powerful reasoning models exhibit minimal or negative performance gains from self-critique compared to standard majority voting. This reveals a consistent limitation in models' self-critique abilities, regardless of scale or reasoning capabilities. In contrast, RefCritic consistently improves performance across nearly all settings, even when supervising larger, more capable reasoning models. With 32 samples, RefCritic supervision improves QwQ performance by 1.5% over majority voting and 1.1% over self-critique. Similar patterns occur for DeepSeek-Distill-Qwen-32B and Qwen 2.5-72B, confirming RefCritic's benefits across model families and scales.

## 6 CONCLUSION

In this work, we introduced RefCritic, a novel approach for training critic models to critique the correctness of solutions and provide effective refinement feedback from LLMs. Our method leverages a dual-reward system that jointly optimizes for judgment accuracy and refinement effectiveness, creating an explicit feedback loop between critique quality and policy model improvement. Our experiments demonstrated that while SFT alone is insufficient for producing comprehensive critiques despite generating better critiques, the integration of reinforcement learning with our designed reward signals significantly enhances both the analytical depth and practical utility of critiques. Experimental results across challenging mathematical datasets and out-of-distribution benchmarks validate RefCritic's effectiveness in consistently enhancing policy model performance in both critique-refinement and majority vote settings. Further experiments on ProcessBench demonstrate that even without a step-level signal, RefCritic can effectively identify the error step.

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

# A  THE USE OF LARGE LANGUAGE MODELS

We utilized LLMs to aid and polish writing.

# B  ABLATION

We propose some ablation studies to understand the role of the two RL training stages in RefCritic, namely $\lambda=0$ and $\lambda=1$. Considering the training cost, we only conduct experiments on RefCritic-Qwen-14B, and ablations on DeepSeek-Distill-Qwen-14B will be added in future research. Specifically, we aim to explore the importance of Refinement Reward. To this end, we mainly compared two groups of experiments: 1) $RL_{\lambda=0}$ and $RL_{\lambda=1}$. 2) $RL_{\lambda=0} \xrightarrow{after} RL_{\lambda=0}$ and $RL_{\lambda=0} \xrightarrow{after} RL_{\lambda=1}$. Each group of experiments is optimized with the same parameters. The results of all these ablation experiments are shown in Table 5. As expected, under the same settings, refinement reward improves the refinement performance of models. Furthermore, first using $RL_{\lambda=0}$ for Critic optimization is also beneficial to RefCritic. $RL_{\lambda=0}$ can quickly improve Critic performance at a lower cost, making $RL_{\lambda=0} \xrightarrow{after} RL_{\lambda=1}$ a setting that balances cost and performance.

| Model | AIME25 $Pass_r@1$ | AIME24 $Pass_r@1$ | Olympiad $Pass_r@1$ |
|---|---|---|---|
| Qwen-14B as Critic | 14.5 | 13.7 | 45.8 |
| *RefCritic-Qwen-14B* | | | |
| $SFT$ | 15.0 | 15.2 | 46.6 |
| $RL_{\lambda=0}$ | 18.5 | 19.1 | 51.4 |
| $RL_{\lambda=1}$ | 19.5 | 21.4 | 54.3 |
| $RL_{\lambda=0} \xrightarrow{after} RL_{\lambda=0}$ | 19.6 | 21.7 | 53.6 |
| $RL_{\lambda=0} \xrightarrow{after} RL_{\lambda=1}$ | **21.2** | **23.0** | **55.7** |

Table 5: Ablation results on RefCritic-Qwen-14B.

# C  DATA CONSTRUCTION

We filtered about 120k problems from the 900k mathematical problems of NuminaMath-1.5 LI et al. (2024). Detailed filter process and utilization can be found in the section C. Our training data pipeline involves rigorous filtering to ensure high-quality and diverse mathematical problems.

**Problem Deduplication**  We start with a deduplication process on the 900k mathematical problems from NuminaMath-1.5 LI et al. (2024). The deduplication process includes the string-based process by performing exact matching after removing special characters such as '\$', '[', ']', etc., and semantic deduplication, where we used gte-multilingual-base embeddings to compute cosine similarity between problem pairs and removing those with similarity scores exceeding 0.95.

**Problem Filter**  Then, we utilize Qwen2.5-72B-Instruct as a judge to filter problems based on several criteria: unsolvable problems, proof problems requiring formal mathematical proofs, and multiple-choice problems. We sample eight solutions for each problem with DeepSeek-Distill-Qwen-7B to ensure appropriate difficulty distribution. We remove problems where DeepSeek-Distill-Qwen-7B either solves all eight attempts correctly or fails on all eight attempts, thus eliminating trivial or impossibly difficult issues. After this comprehensive filtering process, we obtain approximately 120k high-quality mathematical problems for training. All prompt templates are provided in the Appendix.

**Solution Sampling**  To create training data for critic models, we sample 8 responses from the policy model for each problem, remove problems where all solutions are correct or incorrect, filter out incomplete generations, and ensure balanced training by retaining at most two responses per problem (one correct and one incorrect). For efficient scaling, all responses are sampled by sglang inference services[4].

---

[4]https://github.com/sgl-project/sglang

| Model | AIME25 | | | AIME24 | | | Olympiad | | |
|---|---|---|---|---|---|---|---|---|---|
| | $Pass_r$@1 | $Maj_c$@8 | $Maj_c$@64 | $Pass_r$@1 | $Maj_c$@8 | $Maj_c$@64 | $Pass_r$@1 | $Maj_c$@8 | $Maj_c$@16 |
| *Qwen-14B as policy model* | | | | | | | | | |
| Qwen-14B Majority Vote *(No Critic)* | 14.4 | 19.2 | 23.3 | 13.7 | 16.5 | 16.6 | 45.8 | 52.2 | 53.6 |
| Qwen-14B Self-Critic | 14.5 | 19.1 | 22.7 | 13.7 | 18.5 | 21.2 | 45.8 | 52.3 | 54.0 |
| DeepCritic-7B (Yang et al., 2025b) | 13.0 | 17.4 | 19.4 | 13.4 | 18.8 | 23.7 | 43.6 | 51.7 | 52.7 |
| ThinkPRM-14B (Khalifa et al., 2025) | 16.1 | 20.1 | 22.3 | 16.8 | 20.7 | **26.9** | 49.7 | 54.7 | 56.6 |
| RefCritic-Qwen-14B(Ours) | | | | | | | | | |
| $\quad SFT$ | 15.0 | 19.3 | 23.4 | 15.2 | 19.2 | 23.9 | 46.6 | 52.5 | 54.3 |
| $\quad RL_{\lambda=0}$ | 18.5 | 20.8 | 22.4 | 19.1 | 20.5 | 23.8 | 51.4 | 55.4 | 57.4 |
| $\quad RL_{\lambda=0} \xrightarrow{after} RL_{\lambda=1}$ | **21.2** | **21.5** | **24.4** | **23.0** | **21.4** | 26.6 | **55.7** | **57.3** | **59.2** |
| *R1-Qwen-14B as policy model* | | | | | | | | | |
| R1-Qwen-14B Majority Vote *(No Critic)* | 49.1 | 61.6 | 62.0 | 67.6 | 78.7 | 80.1 | 77.7 | 82.7 | 83.3 |
| R1-Qwen-14B Self-Critic | 50.0 | 60.6 | 62.9 | 70.5 | 79.3 | 82.4 | 78.8 | 82.7 | 83.3 |
| DeepCritic (Yang et al., 2025b) | 49.1 | 58.0 | 59.0 | 67.3 | 77.2 | 78.8 | 76.1 | 81.6 | 82.1 |
| ThinkPRM-14B (Khalifa et al., 2025) | 43.9 | 58.7 | 61.2 | 62.6 | 76.3 | 81.0 | 75.2 | 82.1 | 82.7 |
| RefCritic-R1-14B(Ours) | | | | | | | | | |
| $\quad SFT$ | 51.3 | 61.6 | 62.8 | 71.4 | 79.4 | **83.1** | 78.7 | 83.0 | 84.4 |
| $\quad RL_{\lambda=0}$ | 55.1 | 64.2 | 67.1 | **73.5** | 80.4 | 82.8 | **80.4** | 83.8 | 84.5 |
| $\quad RL_{\lambda=0} \xrightarrow{after} RL_{\lambda=1}$ | **56.3** | **65.2** | **68.1** | 72.8 | 80.4 | 82.5 | 80.3 | **83.9** | **84.7** |

Table 6: Detailed performance comparison of different approaches on AIME24/25 and Olympiad.

## D  DETAILED MAIN RESULTS

Due to space constraints in the main text, we did not present the complete performance results of the two-stage RL training in Table 2. Table 6 shows the complete RL training performance. Please note that $RL_{\lambda=1}$ indicates RL with Refinement Feedback. Considering the cost of sampling refinements, we initially set $\lambda = 0$ to achieve rapid improvement in critic performance.

## E  COMPUTATIONAL COST ANALYSIS

Please note that we employ two strategies to control training costs:

1. RefCritic only performs Refinement on solutions that are both incorrect and judged as incorrect by the Critic.

2. Refinement Reward is not used during the initial phase of reinforcement learning.

**Symbol Definitions:**

| Symbol | Definition |
|---|---|
| $c_c$ | Cost of a single Critic rollout during GRPO training |
| $c_r$ | Cost of a single Refinement rollout during GRPO training |
| $r_c$ | Number of Critique rollouts per solution |
| $r_r$ | Number of Refinement rollouts per solution |
| $a$ | RefCritic's judgment accuracy on incorrect solutions |
| $m$ | Number of training steps without Refinement Reward |
| $n$ | Number of training steps with Refinement Reward |

Assuming both standard GRPO and RefCritic are trained for $m + n$ steps:

- **Cost of standard GRPO:** $Cost_{GRPO} = (m + n)r_c c_c$

- **Cost of RefCritic (Phase 1, m steps, no Refinement Reward):** $mr_c c_c$

- **Cost of RefCritic (Phase 2, n steps, with Refinement Reward):** $nr_c c_c + (an/2)r_c r_r c_r$

  $nr_c c_c$ represents the cost of critique rollouts, and the second term represents the additional refinement cost. Since we maintain a balanced RL dataset with equal proportions of correct and incorrect solutions, there are $an/2$ samples requiring refinement rollouts. Thus, the actual refinement cost is $anr_c r_r c_r/2$.

The ratio $T$ of additional refinement cost to original cost (only critique cost) is:

$$T = \frac{anr_c r_r c_r}{2(m+n)r_c c_c} = \frac{anr_r c_r}{2(m+n)c_c} = a\frac{n}{2m+2n}r_r\frac{c_r}{c_c}$$

In our experiments, we set $m : n = 2 : 1$ and $r_r = 8$. Based on empirical observation, $a$ can be approximated as 0.8. Thus:

$$T = 0.8 \times \frac{1}{6} \times 8 \times \frac{c_r}{c_c} = \frac{16}{15} \times \frac{c_r}{c_c}$$

Regarding $\frac{c_r}{c_c}$, we can reasonably assume it is close to 1. As noted in Section 5.4 CRITIC PERFORMANCE, the average output length of RefCritic-Qwen increased from approximately 500 to 3500 tokens, while RefCritic-R1 increased from 3000 to 8000 tokens. Besides, we do not provide the full Critique output to the Policy Model for Refinement. Given that the Critic's generation length is substantially longer than the Policy Model's Refinement, the actual value of this ratio should be lower.

Thus, we can conclude:

$$T \approx 1$$

## F  CASE STUDY

Below is a case study of a critic with and without a refinement reward. The critic with refinement reward finds the issue of the original solution and provides valuable guidance for the policy model to refine the solution. The critic without refinement, in contrast, only provides a critique that does not point out the specific issue of the original solution, leading to an incorrect refinement.

---

**A Case Study of Critique from Critic with and without Refinement Reward**

**PROBLEM**

The sequence $2, 5, 10, 50, 500, ...$ is formed so that each term after the second is the product of the two previous terms. The 15 th term ends with exactly $k$ zeroes. What is the value of $k$ ?

ANSWER: 233

**Original Solution**

To solve the problem, we need to understand the pattern in the sequence and how the number of trailing zeroes in each term evolves.

The sequence is given as (2, 5, 10, 50, 500, ...), and each term after the second is the product of the two previous terms. Let's denote the terms of the sequence as $(a_1, a_2, a_3, ...)$.

Starting with the initial terms: $[a_1 = 2,]$
$[a_2 = 5,]$
$[a_3 = a_1 * a_2 = 2 * 5 = 10,]$
$[a_4 = a_2 * a_3 = 5 * 10 = 50,]$
$[a_5 = a_3 * a_4 = 10 * 50 = 500,]$
$[a_6 = a_4 * a_5 = 50 * 500 = 25000,]$
$[a_7 = a_5 * a_6 = 500 * 25000 = 12500000,]$
$[a_8 = a_6 * a_7 = 25000 * 12500000 = 312500000000,]$
$[a_9 = a_7 * a_8 = 12500000 * 312500000000 = 3906250000000000000,]$
$[a_{10} = a_8 * a_9 = 312500000000 * 3906250000000000000 = 1220703125000000000000000000000,]$

We can observe that the number of trailing zeroes in each term is increasing. To find a pattern, let's look at the number of trailing zeroes in each term:

$(a_1 = 2)(0 trailing zeroes)$
$(a_2 = 5)(0 trailing zeroes)$
$(a_3 = 10)(1 trailing zero)$
$(a_4 = 50)(1 trailing zero)$
$(a_5 = 500)(2 trailing zeroes)$
$(a_6 = 25000)(3 trailing zeroes)$
$(a_7 = 12500000)(6 trailing zeroes)$
$(a_8 = 312500000000)(11 trailing zeroes)$
$(a_9 = 3906250000000000000)(18 trailing zeroes)$

$(a_{10} = 12207031250000000000000000000)(27 trailing zeroes)$

The number of trailing zeroes in each term can be described by the formula:

$$[z_n = \left\lfloor \frac{n-1}{2} \right\rfloor + \left\lfloor \frac{n-2}{2} \right\rfloor + \left\lfloor \frac{n-3}{2} \right\rfloor + \cdots + \left\lfloor \frac{1}{2} \right\rfloor + 0]$$

For (n = 15): $[z_{15} = \left\lfloor \frac{14}{2} \right\rfloor + \left\lfloor \frac{13}{2} \right\rfloor + \left\lfloor \frac{12}{2} \right\rfloor + \cdots + \left\lfloor \frac{1}{2} \right\rfloor + 0]$

$[z_{15} = 7 + 6 + 6 + 5 + 5 + 4 + 4 + 3 + 3 + 2 + 2 + 1 + 1 + 0 + 0]$

$[z_{15} = 56]$

Therefore, the number of trailing zeroes in the 15th term is ( $\boxed{56}$ ).

### Critique without Refinement Reward

...(Long Chain of Thought Analysis)

**Correctness**: Incorrect

**Comment**: The formula the student used is correct, but the final sum of the series was computed incorrectly. The sum of the floor function from $\left\lfloor \frac{14}{2} \right\rfloor$ to $\left\lfloor \frac{1}{2} \right\rfloor$ is **55**, not **56**. Be careful when summing sequences like this—double-check the arithmetic and verify each term step-by-step.

### Refinement Solution based on Critique without Refinement Reward

To address the teacher's feedback and ensure the accuracy of the solution, let's carefully re-evaluate the sum of the series step-by-step.

Given the formula: $[z_n = \left\lfloor \frac{n-1}{2} \right\rfloor + \left\lfloor \frac{n-2}{2} \right\rfloor + \left\lfloor \frac{n-3}{2} \right\rfloor + \cdots + \left\lfloor \frac{1}{2} \right\rfloor + 0]$

For ( n = 15 ): $[z_{15} = \left\lfloor \frac{14}{2} \right\rfloor + \left\lfloor \frac{13}{2} \right\rfloor + \left\lfloor \frac{12}{2} \right\rfloor + \cdots + \left\lfloor \frac{1}{2} \right\rfloor + 0]$

We compute each term individually: $\left\lfloor \frac{14}{2} \right\rfloor = 7$

$\left\lfloor \frac{13}{2} \right\rfloor = 6$

$\left\lfloor \frac{12}{2} \right\rfloor = 6$

$\left\lfloor \frac{11}{2} \right\rfloor = 5$

$\left\lfloor \frac{10}{2} \right\rfloor = 5$

$\left\lfloor \frac{9}{2} \right\rfloor = 4$

$\left\lfloor \frac{8}{2} \right\rfloor = 4$

$\left\lfloor \frac{7}{2} \right\rfloor = 3$

$\left\lfloor \frac{6}{2} \right\rfloor = 3$

$\left\lfloor \frac{5}{2} \right\rfloor = 2$

$\left\lfloor \frac{4}{2} \right\rfloor = 2$

$\left\lfloor \frac{3}{2} \right\rfloor = 1$

$\left\lfloor \frac{2}{2} \right\rfloor = 1$

$\left\lfloor \frac{1}{2} \right\rfloor = 0$

Now, sum these values: $[7 + 6 + 6 + 5 + 5 + 4 + 4 + 3 + 3 + 2 + 2 + 1 + 1 + 0 + 0]$

Let's verify the addition step-by-step: 7 + 6 = 13

13 + 6 = 19

19 + 5 = 24

24 + 5 = 29

29 + 4 = 33

33 + 4 = 37

37 + 3 = 40

40 + 3 = 43

43 + 2 = 45

45 + 2 = 47

47 + 1 = 48

48 + 1 = 49

49 + 0 = 49

49 + 0 = 49

The correct sum is: 49

Thus, the number of trailing zeroes in the 15th term is ( 49 ).

**Critique with Refinement Reward**

...(Long Chain of Thought Analysis)

**Correctness**: Incorrect

**Comment**: The number of trailing zeroes in the 15th term is not given by the formula $\sum_{k=1}^{n-1} \lfloor \frac{k}{2} \rfloor$. Instead, it is determined by the minimum of the number of factors of 2 and 5 in the prime factorization of the term, which follows a recurrence based on the sequence definition. You should trace the number of factors of 2 and 5 in each term using the recurrence relation $a_n = a_{n-1} * a_{n-2}$, and then compute the number of trailing zeroes as $\min(p_2(n), p_5(n))$.

**Refinement Solution based on Critique with Refinement Reward**

To find the number of trailing zeroes in the 15th term of the sequence, we need to focus on the number of factors of 2 and 5 in each term. Given the sequence definition, each term is the product of the two preceding terms, and we need to determine how these factors evolve. Let's denote the number of factors of 2 in $a_n$ as $p_2(n)$ and the number of factors of 5 in $a_n$ as $p_5(n)$.

Starting with the initial terms:

$$[a_1 = 2 \Rightarrow p_2(1) = 1, \quad p_5(1) = 0,]$$
$$a_2 = 5 \Rightarrow p_2(2) = 0, \quad p_5(2) = 1.$$

For the subsequent terms, we use the recurrence relation:

$$[a_n = a_{n-1} * a_{n-2} \Rightarrow p_2(n) = p_2(n-1) + p_2(n-2) \text{ and } p_5(n) = p_5(n-1) + p_5(n-2).]$$

We calculate the values for $(p_2(n)) and (p_5(n)) upto (n = 15)$ :

| $n$ | $p_2(n)$ | $p_5(n)$ |
|-----|----------|----------|
| 1 | 1 | 0 |
| 2 | 0 | 1 |
| 3 | 1 | 1 |
| 4 | 1 | 2 |
| 5 | 2 | 3 |
| 6 | 3 | 5 |
| 7 | 5 | 8 |
| 8 | 8 | 13 |
| 9 | 13 | 21 |
| 10 | 21 | 34 |
| 11 | 34 | 55 |
| 12 | 55 | 89 |
| 13 | 89 | 144 |
| 14 | 144 | 233 |
| 15 | 233 | 377 |

The number of trailing zeroes in the 15th term is determined by the minimum of $(p_2(15))$ and $(p_5(15))$: $[\min(p_2(15), p_5(15)) = \min(233, 377) = 233.]$
Thus, the number of trailing zeroes in the 15th term is ( 233 ).

## G    TEMPLATES

The template we used for critique (Qwen based critic)

Given a student's solution, analyze it step-by-step to determine correctness. Do not solve the problem yourself, provide feedback focus on the student's work to help them learn. First, analyze the student's solution step by step, then conclude your feedback. Output your critique as:

### Analysis
Step wise analysis of the student's solution.

### Feedback
**Correctness**: Correct/Incorrect
(If incorrect)
**Comment**: Identify the specific error in the solution and help the student recognize why their approach leads to an incorrect result. Then, provide a comment that will help the student to resolve this problem. Do not expose any answer!

[Problem]
{problem}

[Solution]
{solution}

---

**The template we used for critique (R1-Distill-Qwen based critic)**

Given a student's mathematical solution, analyze it step-by-step to determine correctness. Do not solve the problem yourself, provide feedback focus on the student's work to help them learn. Conclude your feedback as:

**Correctness**: Correct / Incorrect
(If incorrect)
**Comment**: Identify the specific error in the solution and help the student recognize why their approach leads to an incorrect result. Then, provide a comment that will help the student to resolve this problem.

Do not expose any answer!

[Problem]
{problem}

[Solution]
{solution}

---

**The template we used for refinement**

Review your solution to a mathematical problem and a feedback from your teacher. Create an improved version that fixes the identified errors. Please reason step by step and put your final answer within □.

[Problem]
{problem}

[Original Solution]
{solution}

[Teacher Feedback]
{critique}

---

**The template we used for evaluating processbench**

The following is a math problem and a solution (split into paragraphs, enclosed with tags and indexed from 0):

[Math Problem]
{problem}

[Solution]
{solution}

Your task is to review and critique the solution paragraph by paragraph. Once you identify an error in a paragraph, return the index of the paragraph where the earliest error occurs. Otherwise, return the index of -1 (which typically denotes "not found").

Please put your final answer (i.e., the index) in ☐.

---

**The template we used to extract the answer of critiques in evaluation with ProcessBench**

[Task]
Given a critique of a multi-step solution to a math problem.

Your job is to identify the index (starts from 0) of the step where the first crucial error occurs according to the critique. Otherwise, output the index of -1 (which typically denotes "solution is correct").

First analyze the critique and solution, then output the index of the step where the first crucial error occurs according to the critique.

Output in the following format:
**Analysis**: {the analysis of the critique}
**Error Index**: {index}

[Critique]
{critique}

Now analyze the solution and critique.

---

