# OpenReview forum: "RefCritic: Training Long Chain-of-Thought Critic Models with Refinement Feedback"
_ICLR.cc/2026/Conference — Submitted to ICLR 2026_

### Official Review · Reviewer_wJJH · 2025-10-31

**Soundness:** 2
**Presentation:** 3
**Contribution:** 2
**Rating:** 2
**Confidence:** 4

**Summary:**

The paper proposes RefCritic, a critic model trained to both (i) judge whether a solution is correct and (ii) provide feedback that effectively helps a policy model improve its answer. The system adopts a two-stage training pipeline: a cold-start SFT on filtered critique data, followed by RL (GRPO) with dual rewards that explicitly align critique generation with downstream solution improvement. Experiments using two 14B critic models trained on math data show consistent improvements in the generator’s self-refined accuracy and majority-vote performance. Further evaluation on ProcessBench demonstrates that RefCritic surpasses several existing baselines. The paper also reports out-of-domain gains on LiveCodeBench and GPQA, as well as the effectiveness of using the RefCritic-14B model to critique stronger models, with varying degrees of improvement.

**Strengths:**

1. **Goal-driven formulation.** The dual-reward scheme explicitly ties critique usefulness to the generator’s performance improvement, addressing a common gap in prior “LLM-as-critic” methods that focus only on correctness. The λ-scheduled reward balancing within the GRPO framework is conceptually simple, computationally efficient, and compatible with most existing RLVR approaches.
2. **Consistent empirical gains.** RefCritic achieves steady improvements in critique-based self-refinement Pass@1 and majority-vote scores across AIME24, AIME25, and OlympiadBench, with RL-finetuned models outperforming their SFT counterparts in most cases. These results validate the effectiveness of the dual-reward algorithm design.
3. **Generalizable performance.** The trained RefCritic-14B further enhances the performance of stronger external models, showing clear gains in 2 out of 3 tested cases. This suggests that the learned critique behavior is generalizable and can assist other models rather than overfitting to the generator-specific style seen in the training. Additional experiments on LiveCodeBench and GPQA confirm the cross-domain applicability of RefCritic beyond mathematical problems.

**Weaknesses:**

1. **[Significance and Novelty]** The idea of training a critic model using both correctness judgments and refinement outcomes has been previously explored, notably in [1]. Several findings and design choices in this paper, such as the limited benefit of SFT-only critics and the generalization to other generators and tasks, are also reported there. While RefCritic focuses on mathematical reasoning rather than coding, the overlap in methodology and conclusions reduces the originality and significance of this work. Clarifying key distinctions or introducing new experiment insights would strengthen the impact of this work.
2. **[Soundness]** (1) The paper omits direct comparison against existing critic models (e.g., those in Table 4) in the main results (Table 2), making it difficult to evaluate the advantage of RefCritic relative to prior methods. (2) In Table 4, the authors deviate from the evaluation protocol in [2], which asks the model to directly output the first erroneous step, by using Qwen2.5-14B-Instruct to extract the answer. This additional extraction step may introduce bias or error, and the derived results are not strictly comparable to those in the ProcessBench. (3) In Figure 3(a), the generator used (Qwen2.5-72B) appears to be a base model without instruction-following ability, which could disadvantage methods like self-critique that rely on instruction comprehension. Using an instruct-tuned generator would yield a fairer comparison.
3. **[Presentation]** Table 2 is somewhat confusing and could be reorganized for clarity. The meaning of entries such as “R1-Qwen-14B Maj” under “Pass_r@1” is not immediately clear. The λ = 0 and λ = 1 configurations could be moved to an ablation section, allowing the main table to focus on key results and improve readability.

[1] Teaching Language Models to Critique via Reinforcement Learning, ICML 2025.

[2] ProcessBench: Identifying Process Errors in Mathematical Reasoning, ACL 2025.

**Questions:**

1. Why are the critic models evaluated on *ProcessBench* (Table 4) not included for comparison in the main experiment (Table 2)?
2. Have the authors explored multi-turn self-refinement using the trained RefCritic models? It would be interesting to see whether iterative critique–refinement cycles can further improve solution accuracy.

---

> ### Author Response · Authors · 2025-11-22
> **Response to Reviewer wJJH (Part 1/3)**
>
> > **W1. The idea of training a critic model using both correctness judgments and refinement outcomes has been previously explored, notably in CTRL...**
> ---
>
> Thank you for highlighting **CTRL** as an important related work. We acknowledge CTRL's pioneering contributions to training critic models, and we have cited and discussed this work in the revised PDF.
>
> However, we must emphasize that RefCritic is not merely a transplantation of CTRL's approach to the mathematical domain. These two approaches differ substantially in several key aspects:
>
> ---
>
> **1. Optimization Objectives: Single-Objective vs. Dual-Objective**\
> CTRL's objective is to maximize the pass rate of refined code, treating the Critic as an auxiliary model to help the Policy Model succeed, without explicit supervision on judgment accuracy. The Critic only needs to produce feedback that leads to passing solutions, without being required to identify correctness genuinely.
>
> RefCritic, by contrast, decouples Judgment Accuracy from Refinement Effectiveness via a dual-reward mechanism. This ensures the Critic must first be a qualified Judge before becoming a good Guide, which offers several key advantages: 1) Judgment is inherently simpler than generating effective feedback for refinement. For Long Reasoning Models with strong base performance, unconditional refinement may introduce noise or corrupt correct answers. By judging correctness first, RefCritic prevents unnecessary interference with correct solutions—a critical consideration when the policy model already achieves high accuracy. 2) RefCritic's explicit judgment capability enables **weighted majority voting** among multiple solutions in parallel, filtering out incorrect candidates rather than requiring serial iterative refinement. As shown in Table 2, RefCritic demonstrates consistent improvements in majority voting scenarios.  3) Conditional refinement avoids "over-refining" valid solutions, significantly reducing RL training costs (detailed in Appendix E).
>
> ---
>
> **2. Expanded Empirical Insights**\
> RefCritic provides two empirical insights into RL-trained critics:
> 1. We demonstrate that RLVR using only outcome-level rewards can supervise models to learn fine-grained, step-level critical abilities.(Section 5.4)
> 2. Although RL training is performed exclusively on math problems, RefCritic transfers to out-of-distribution domains such as Code (LiveCodeBench) and Science (GPQA), where it yields substantially larger gains than a self-critique baseline(Section 5.3). This indicates that RefCritic goes beyond a simple math discriminator and instead learns a more general “critique–refinement” strategy.
>
> ---
>
> **3. Difference in Critique Form**
> CTRL trains the Critic to generate structured feedback text with all content returned to the policy model as external signals for refinement. This may cause the Critic's own biases to be transmitted to the policy model through the critique.
>
> RefCritic instead separates the critic’s internal reasoning from the external feedback: we train a long chain-of-thought critic whose analysis can be rich and detailed, while the exposed feedback remains short and focused (Appendix G). Through RL, RefCritic's average reasoning length increases significantly, and this growth occurs primarily in the critique reasoning. This internalization of the long chain-of-thought not only enables RefCritic to emerge with the capability to precisely localize erroneous steps without any step-level supervision, but also empowers the model to demonstrate superior capability when critiquing stronger models (Section 5.6).
>
> ---
>
> In summary, although RefCritic and CTRL share high-level similarities, they differ in practice. We will cite and discuss CTRL in the Related Work section of the revised version. We thank the reviewer for helping us clarify RefCritic's academic contributions.

---

> ### Author Response · Authors · 2025-11-22
> **Response to Reviewer wJJH (Part 2/3)**
>
> > **W2.[Soundness(1)]&Q1. The paper omits direct comparison against existing critic models (e.g., those in Table 4) in the main results (Table 2), making it difficult to evaluate the advantage of RefCritic relative to prior methods.**
> ---
>
> Thank you for your suggestion. We have added comparisons with other critic models from Table 4 to the main results. We exclude SCRIT-72B from the comparison as its weights are not open-sourced. The specific performance is as follows:
>
> Qwen-14B as policy model
> | Model | AIME25 $Pass_r@1$ | AIME25 $Maj_c@8$ | AIME25 $Maj_c@64$ | AIME24 $Pass_r@1$ | AIME24 $Maj_c@8$ | AIME24 $Maj_c@64$ | Olympiad $Pass_r@1$ | Olympiad $Maj_c@8$ | Olympiad $Maj_c@16$ |
> | :--- | :---: | :---: | :---: | :---: | :---: | :---: | :---: | :---: | :---: |
> | Qwen-14B Majority Vote *(No Critic)* | 14.4 | 19.2 | 23.3 | 13.7 | 16.5 | 16.6 | 45.8 | 52.2 | 53.6 |
> | Qwen-14B Self-Critic | 14.5 | 19.1 | 22.7 | 13.7 | 18.5 | 21.2 | 45.8 | 52.3 | 54.0 |
> | DeepCritic-7B | 13.0 | 17.4 | 19.4 | 13.4 | 18.8 | 23.7 | 43.6 | 51.7 | 52.7 |
> | ThinkPRM-14B | 16.1 | 20.1 | 22.3 | 16.8 | 20.7 | **26.9** | 49.7 | 54.7 | 56.6 |
> | RefCritic-Qwen-14B(SFT) | 15.0 | 19.3 | 23.4 | 15.2 | 19.2 | 23.9 | 46.6 | 52.5 | 54.3 |
> | RefCritic-Qwen-14B(RL) | **21.2** | **21.5** | **24.4** | **23.0** | **21.4** | 26.6 | **55.7** | **57.3** | **59.2** |
>
> R1-Qwen-14B as policy model
> | Model | AIME25 $Pass_r@1$ | AIME25 $Maj_c@8$ | AIME25 $Maj_c@64$ | AIME24 $Pass_r@1$ | AIME24 $Maj_c@8$ | AIME24 $Maj_c@64$ | Olympiad $Pass_r@1$ | Olympiad $Maj_c@8$ | Olympiad $Maj_c@16$ |
> | :--- | :---: | :---: | :---: | :---: | :---: | :---: | :---: | :---: | :---: |
> | R1-Qwen-14B Majority Vote *(No Critic)* | 49.1 | 61.6 | 62.0 | 67.6 | 78.7 | 80.1 | 77.7 | 82.7 | 83.3 |
> | R1-Qwen-14B Self-Critic | 50.0 | 60.6 | 62.9 | 70.5 | 79.3 | 82.4 | 78.8 | 82.7 | 83.3 |
> | DeepCritic | 49.1 | 58.0 | 59.0 | 67.3 | 77.2 | 78.8 | 76.1 | 81.6 | 82.1 |
> | ThinkPRM-14B | 43.9 | 58.7 | 61.2 | 62.6 | 76.3 | 81.0 | 75.2 | 82.1 | 82.7 |
> | RefCritic-R1-14B(SFT) | 51.3 | 61.6 | 62.8 | 71.4 | 79.4 | **83.1** | 78.7 | 83.0 | 84.4 |
> | RefCritic-R1-14B(RL) | **56.3** | **65.2** | **68.1** | **72.8** | **80.4** | 82.5 | **80.3** | **83.9** | **84.7** |
>
> ---
>
> Experiments demonstrate that RefCritic not only outperforms all baselines in fine-grained critique tasks but also exhibits superior performance when interacting with the Policy Model. We will include these results in Section 5.2: MAIN RESULTS of the revised PDF.
>
> ---
> > **W2.[Soundness(2)]. In Table 4, the authors deviate from the evaluation protocol in [2], which asks the model to directly output the first erroneous step, by using Qwen2.5-14B-Instruct to extract the answer. This additional extraction step may introduce bias or error, and the derived results are not strictly comparable to those in the ProcessBench.**
> ---
>
> Thank you for your careful review. Regarding our design choice to use Qwen2.5-14B-Instruct for answer extraction, we offer the following response:
> 1. Unlike DeepCritic or other works, RefCritic is trained end-to-end without using step-level reward signals. Therefore, ProcessBench's objective of "identifying the first error step" is an Out-of-Distribution task for RefCritic. However, enhanced by long-chain-of-thought Critique after two-stage RL training, RefCritic can still identify accurate error steps. The only additional intervention required is extracting the specific step index.
> 2. As noted in Table 4, Qwen14B itself has relatively low performance (only 49.7), whereas RefCritic's performance is significantly higher. Thus, using Qwen14B as an extractor would not lead to an overestimation of RefCritic's performance.
> 3. Through manual evaluation of 50 randomly sampled cases, we found Qwen14B's extraction accuracy to be 96%. Among the two error cases, one involved RefCritic finding the accurate error step, but Qwen14B failing to extract it correctly (RefCritic identified two error steps, and Qwen14B incorrectly extracted the index of the second error step).
>
> Therefore, using Qwen14B as an extractor does not overestimate RefCritic's performance and may even underestimate it.
>
> ---
> > **W2.[Soundness(3)]. In Figure 3(a), the generator used (Qwen2.5-72B) appears to be a base model without instruction-following ability, which could disadvantage methods like self-critique that rely on instruction comprehension. Using an instruct-tuned generator would yield a fairer comparison.**
> ---
>
> The model we used is **Qwen2.5-72B-Instruct**, not the base model Qwen2.5-72B. All models used in this paper possess instruction-following capabilities. We will revise this description to clarify the experimental details. You can view our changes in the newly revised PDF.

---

> ### Author Response · Authors · 2025-11-22
> **Response to Reviewer wJJH (Part 3/3)**
>
> > **W3. Table 2 is somewhat confusing and could be reorganized for clarity. The meaning of entries such as “R1-Qwen-14B Maj” under “Pass_r@1” is not immediately clear. The λ = 0 and λ = 1 configurations could be moved to an ablation section, allowing the main table to focus on key results and improve readability.**
> ---
>
> Thank you for your suggestion. $RL_{\lambda=0}$ and $RL_{\lambda=0} \rightarrow RL_{\lambda=1}$ represent the performance of RefCritic's two-stage reinforcement learning. The former indicates performance after training without the Refinement Reward, while the latter indicates performance after further training with the Refinement Reward. We will omit the separate performance of $RL_{\lambda=0}$ in the main experiments and present it only in Appendix B and Appendix C.
>
> Regarding "R1-Qwen-14B Maj," as mentioned in the table caption, **Qwen-14B Maj** and **R1-Qwen-14B Maj** denote the baseline that performs majority vote directly with related calculations. We will reorganize the table format to avoid misunderstanding. You can view our changes in the revised PDF.
>
> ---
> > **Q2. Have the authors explored multi-turn self-refinement using the trained RefCritic models? It would be interesting to see whether iterative critique–refinement cycles can further improve solution accuracy.**
> ---
>
> Thank you very much for your suggestion. This experiment indeed led to some interesting results. We will add this part of the experiment to the revised PDF.
>
> For each solution, we iteratively perform critique and refinement until RefCritic judges the Policy Model's solution to be correct, or the maximum iteration limit (8) is reached. We tested RefCritic-Qwen-14B(RL) on three mathematical benchmarks. The testing process is identical to the main experiments. For AIME, we sample 128 initial solutions, then perform iterative Critique and Refinement on 3,840 solutions, and finally report average performance. Olympiad samples only 32 times.
>
> The specific experimental results are as follows:
>
> ---
>
> | Dataset | Round 0 | Round 1 | Round 2 | Round 3 | Round 4 | Round 5 | Round 6 | Round 7 | Round 8 |
> | :--- | :--- | :--- | :--- | :--- | :--- | :--- | :--- | :--- | :--- |
> | **AIME24** | 13.6 | 23.0 | 24.8 | 25.6 | 26.2 | 26.6 | 26.8 | 27.0 | 27.2 |
> | **AIME25** | 14.4 | 21.2 | 22.8 | 23.4 | 23.8 | 24.1 | 24.3 | 24.5 | 24.5 |
> | **Olympiad** | 45.7 | 55.7 | 56.9 | 58.8 | 59.3 | 59.6 | 59.8 | 60.0 | 60.1 |
>
> ---
>
> The experiments show that iterative refinement with the RefCritic model can further enhance performance. Encouragingly, with just 8 rounds of iterative refinement, RefCritic's performance approaches or even surpasses **Maj_c@64** (Majority Vote with Critic). However, this iterative improvement gradually saturates. Through manual inspection, we found that this is because: 1) after several iterations, most problems solvable by the Policy Model have been addressed, and the remaining problems are beyond the model's scope. 2) RefCritic tends to misjudge the correctness of refined solutions that follow its own feedback, which is an interesting problem we may look into in the future.
>
> Relevant results can be found in Section 5.5 TEST-TIME SCALING of the revised PDF.

---

> > ### Comment · Reviewer_wJJH · 2025-11-27
> >
> > Thank you for your detailed response. Your clarifications have addressed my some of my concerns. I have updated the my score.

---

### Official Review · Reviewer_XRcp · 2025-10-31

**Soundness:** 3
**Presentation:** 3
**Contribution:** 3
**Rating:** 4
**Confidence:** 4

**Summary:**

The paper proposes RefCritic, a reinforcement learning (RL)-based framework for training long chain-of-thought (CoT) critic models that provide actionable feedback to improve the reasoning of a policy language model. The core idea is to move beyond supervised fine-tuning (SFT) by introducing a dual-reward RL objective: instance-level correctness and refinement accuracy. The authors train RefCritic on mathematical reasoning tasks using models like Qwen2.5-14B and DeepSeek-R1-Distill-Qwen-14B, and evaluate on benchmarks including AIME24/25, OlympiadBench, and ProcessBench. Results show consistent gains in both refinement-after-critique (e.g., +6.8–7.2% Pass@1 on AIME25) and majority-vote filtering settings.

**Strengths:**

1. The paper is well-written and logically structured. The
2. The experimental design and scale are huge
3. While the dual-reward concept is not entirely novel, the specific formulationis a meaningful operationalization.

**Weaknesses:**

1. The central idea, using refinement performance as a reward signal for training critics, has been explored in prior work. For instance, Training Language Models to Critique With Multi-agent Feedback also leverages feedback loops where critique quality is tied to downstream correction success. The paper would benefit from a more nuanced discussion of these related approaches in Section 2.
2. The method critically relies on binary, verifiable ground truth (e.g., mathematical answers, code execution) that help to provide the accuracy of responses and critiques. This limits its applicability to open-ended, subjective, or real-world tasks common scenarios. The out-of-distribution results on GPQA and LiveCodeBench are encouraging but still within “closed-answer” regimes. The paper does not address how RefCritic would function in settings without deterministic evaluation, which weakens its broader relevance.
3. The refinement reward requires sampling multiple refined solutions per critique during training, which is computationally expensive.

**Questions:**

No question

---

> ### Author Response · Authors · 2025-11-22
> **Response to Reviewer XRcp (Part 1/3)**
>
> > **W1. The central idea, using refinement performance as a reward signal for training critics, has been explored in prior work...**
> ---
>
> We thank the reviewer for highlighting this significant related work. We align with the perspective presented in the *Multi-Agent-Revision-Scoring Filtering* section of MultiCritique, which posits that the value of a critique should be measured by its actual utility in aiding refinement. We will cite and discuss this work in the Related Work section of the revised manuscript. However, despite sharing this high-level philosophy, RefCritic differs fundamentally from MultiCritique in several aspects:
>
> ---
>
> **1. Role of Refinement in the Training Pipeline (Filtering Phase vs. Training Phase)**\
> MultiCritique is a robust data construction and filtering framework. It uses refinement improvement as a filter (i.e., the Multi-Agent-Revision-Scoring mechanism) to select high-quality critique preference pairs. These data are then used to train a separate Reward Model, which subsequently optimizes the Critic.
>
> In contrast, RefCritic incorporates refinement directly into the reward during on-policy RL training. We let the policy model generate multiple on-policy refinements based on the critique and directly use the refinement results' accuracy as the reward. This design establishes a direct loop between critique quality and refinement effectiveness.
>
> ---
>
> **2. Deep Chain-of-Thought Internalization vs. Multi-Agent Aggregation**\
> MultiCritique generates structured Analytical Critique Units, aggregating viewpoints from multiple models to reduce hallucinations. It is a multi-agent aggregation process.
>
> RefCritic focuses on the depth expansion of the chain of thought. Through two-stage reinforcement learning, RefCritic can spontaneously generate analyses spanning thousands of tokens. Crucially, this growth primarily occurs in the critique analysis part rather than the feedback part that will be transmitted to the policy model. This Long CoT Deep Critic enables RefCritic to emerge with step-level error localization capabilities without fine-grained label supervision.
>
> ---
>
> **3. Task Differences (Open-Domain Tasks vs. Reasoning Tasks)**\
> MultiCritique employs `InternLM2-20B-reward` to evaluate the quality of solutions and revisions for open-domain tasks like dialogue and summarization. Although it uses exact answer matching for mathematical problems, this is only employed during the MARS filtering stage.
>
> RefCritic is specifically designed for reasoning tasks with objective ground truths. We explicitly separate two optimization objectives: judgment accuracy and refinement effectiveness. In RefCritic, refinement is considered only when the Critic correctly identifies the errors in the incorrect solutions.
>
> ---
>
> In summary, although RefCritic and MultiCritique share high-level concepts, their research directions and objectives differ significantly.

---

> ### Author Response · Authors · 2025-11-22
> **Response to Reviewer XRcp (Part 2/3)**
>
> > **W2. The method critically relies on binary, verifiable ground truth (e.g., mathematical answers, code execution) to assess the accuracy of responses and critiques...**
> ---
>
> We appreciate your insights and observations regarding generalizability. While the RL training paradigm followed by RefCritic—**RLVR (Reinforcement Learning with Verifiable Rewards)**—leverages the deterministic nature of math/code environments, this does not imply that RLVR is not significant for general domains. We provide the following clarifications:
>
> 1. Recently, some works have focused on feedback in general domains [1, 2], but these studies invariably encounter difficulties such as constructing high-quality preference pairs and reward hacking. RefCritic adopts a dual-reward mechanism using deterministic answers as the reward source to enhance the model's Critic capability, while general domains are not the primary focus of this paper. Moreover, RefCritic's robustness in Out-of-Distribution scenarios (GPQA and LiveCodeBench) demonstrates that it goes beyond simple answer fitting, proving that RefCritic learns a generalizable "critique-refinement" strategy.
>
> 2. Recent research indicates that RLVR can stimulate general process generalization capabilities. Specifically, training in verifiable environments, such as math/code, learns underlying cognitive abilities for problem decomposition, self-correction, and long-term planning. As validated by DeepSeek-R1 [3], these capabilities are generalizable and can be directly transferred to open-ended tasks, demonstrating that ground-truth-based training can significantly enhance the ability to handle open-ended, non-deterministic problems.
>
> 3. Current research is extending RLVR-like methods to open domains through approaches like "Rubrics as Rewards" or Generative Reward Models (GenRM) [4,5]. RefCritic's ability to construct accurate judgments and practical suggestions, trained in high-confidence environments, can be a strong foundation for these broader applications.
>
> [1] Training Language Models to Critique With Multi-agent Feedback\
> [2] HelpSteer3: Human-Annotated Feedback and Edit Data to Empower Inference-Time Scaling in Open-Ended General-Domain Tasks\
> [3] DeepSeek-R1: Incentivizing Reasoning Capability in LLMs via Reinforcement Learning\
> [4] Rubrics as Rewards: Reinforcement Learning Beyond Verifiable Domains\
> [5] RLBFF: Binary Flexible Feedback to bridge between Human Feedback & Verifiable Rewards

---

> ### Author Response · Authors · 2025-11-22
> **Response to Reviewer XRcp (Part 3/3)**
>
> > **W3. The refinement reward requires sampling multiple refined solutions per critique during training, which is computationally expensive.**
> ---
>
> The calculation of the Refinement Reward in RefCritic indeed introduces additional computational costs. However, our calculations show that this cost is not prohibitively high. Specifically, the Refinement Reward incurs only about $1\times$ additional computational cost. In comparison, the use of the Refinement Reward yields an average performance improvement of about 10% across three mathematical benchmarks, as detailed in Appendix B ABLATION. This trade-off is worthwhile. The detailed cost estimation is as follows:
>
> Please note that we employ two strategies to control training costs:
> 1. RefCritic only performs Refinement on solutions that are both incorrect and judged as incorrect by the Critic.
> 2. Refinement Reward is not used during the initial phase of reinforcement learning.
>
> **Symbol Definitions:**
>
> | Symbol | Definition |
> |------|------|
> | $c_c$ | Cost of a single Critic rollout during GRPO training |
> | $c_r$ | Cost of a single Refinement rollout during GRPO training |
> | $r_c$ | Number of Critique rollouts per solution |
> | $r_r$ | Number of Refinement rollouts per solution |
> | $a$ | RefCritic's judgment accuracy on incorrect solutions |
> | $m$ | Number of training steps without Refinement Reward |
> | $n$ | Number of training steps with Refinement Reward |
>
> Assuming both standard GRPO and RefCritic are trained for $m+n$ steps:
> * **Cost of standard GRPO:** $Cost_{GRPO} = (m+n)r_cc_c$
> * **Cost of RefCritic (Phase 1, $m$ steps, no Refinement Reward):** $mr_cc_c$
> * **Cost of RefCritic (Phase 2, $n$ steps, with Refinement Reward):** $nr_cc_c+(an/2)r_cr_rc_r$
>     * $nr_cc_c$ represents the cost of critique rollouts, and the second term represents the additional refinement cost. Since we maintain a balanced RL dataset with equal proportions of correct and incorrect solutions, there are $an/2$ samples requiring refinement rollouts. Thus, the actual refinement cost is $anr_cr_rc_r/2$.
>
> The ratio $T$ of additional refinement cost to original cost (only critique cost) is:
> $$T=\frac{anr_cr_rc_r}{2(m+n)r_cc_c}=\frac{anr_rc_r}{2(m+n)c_c}=a\frac{n}{2m+2n}r_r\frac{c_r}{c_c}$$
>
> In our experiments, we set $m:n=2:1$ and $r_r=8$. Based on empirical observation, $a$ can be approximated as 0.8. Thus:
> $$T=0.8 \times \frac{1}{6} \times 8 \times \frac{c_r}{c_c} = \frac{16}{15} \times \frac{c_r}{c_c}$$
>
> Regarding $\frac{c_r}{c_c}$, we can reasonably assume it is close to 1. As noted in Section 5.4 CRITIC PERFORMANCE, the average output length of RefCritic-Qwen increased from approximately 500 to 3500 tokens, while RefCritic-R1 increased from 3000 to 8000 tokens. Besides, we do not provide the full Critique output to the Policy Model for Refinement (only the feedback part, you can find the exact template in Appendix F and G). Given that the Critic's generation length is substantially longer than the Policy Model's Refinement, the actual value of this ratio should be lower.
>
> Thus, we can conclude:
> $$T \approx 1$$
>
> We will include the detailed cost analysis in Appendix E of the revised PDF.

---

> > ### Comment · Reviewer_XRcp · 2025-11-23
> >
> > Thanks for your response. I think your response largely addresses my concerns. I recommend incorporating the additional information provided in the rebuttal—particularly the discussion of the paper *“Training Language Models to Critique with Multi-Agent Feedback”*—into the revised version of the paper.
> > I am considering raising my score to 6.

---

### Official Review · Reviewer_qP4v · 2025-11-08

**Soundness:** 3
**Presentation:** 3
**Contribution:** 3
**Rating:** 6
**Confidence:** 4

**Summary:**

This paper proposes a LLM critic model based on reinforcement learning (RL) with dual rule-based rewards: (1) instance-level correctness of solution judgments and (2) refinement accuracies of the policy model based on critiques, aiming to generate high-quality evaluations with actionable feedback that effectively guides model refinement. Experimental results on five benchmarks show the effectiveness of the proposed method.

**Strengths:**

1. This paper provides analyses on SFT-based critic models, which gives some meaningful insights.
2. Empirical results show the superior performance of the proposed method.
3. This paper is overall well-organized.

**Weaknesses:**

1. This paper misses an important line of work about critique generation for refinement, such as [1]. In my view, the proposed method is similar to [1], especially the design of R_r (Equation 6). The authors should clearly discuss the difference to highlight their core novelty.

2. Although the authors claim that their method is a long-chain-of-thought critic module, I do not find how this method can improve the long-chain-of-thought generation ability. Now, the methodological design is mainly aimed at better refinement.

3. The quality of generated critiques themselves should be assessed via automatic metrics or human evaluation.

4. The content of Line 262 should be after that of Line 249. The explanation of \lambda should be after the apperance.

[1] Training Language Model to Critique for Better Refinement. ACL 2025 Findings.

**Questions:**

I have included my questions in the weaknesses part.

---

> ### Author Response · Authors · 2025-11-22
> **Response to Reviewer qP4v (Part 1/2)**
>
> > **W1. This paper misses an important line of work about critique generation for refinement, such as [Training Language Model to Critique for Better Refinement.]. In my view, the proposed method is similar to [1], especially the design of R_r (Equation 6). The authors should clearly discuss the difference to highlight their core novelty.**
> ---
>
> Thank you for highlighting this important related work. We will cite and discuss it in the revised PDF. We concur with RCO's core philosophy of utilizing refinement outcomes to measure critique quality. Although both approaches utilize refinement signals to train critics at a high level, RefCritic and RCO differ substantially in several key aspects. These design differences make RefCritic more suitable for high-precision mathematical and logical reasoning tasks, whereas RCO is designed as a general cross-task framework (e.g., dialogue, summarization, QA).
>
> ---
>
> **1. Nature of Reward Signal (Preference-based Reward vs. Objective Ground Truth)**\
> RCO's core metric, *Critique Utility*, relies on a Judge model to compare the Refined Response against the Initial Response. This preference-based relative scoring mechanism is suitable for open-domain tasks, but may introduce additional alignment bias in mathematical reasoning.
>
> RefCritic focuses on reasoning tasks with standard answers. Our Refinement Reward is calculated based on objective correctness. In RefCritic, an effective critique must guide the model to produce the "correct" answer, not a "better" answer.
>
> ---
>
> **2. Reward Composition (Single Reward vs. Dual Rewards)**\
> RCO employs a single Critique Utility as the optimization objective, focusing on the refinement improvement. It does not explicitly distinguish between the Critic's judgment accuracy and the effectiveness of its guidance. It is not designed to train a critic that determines correctness, but rather one that focuses on generating improvement feedback.
>
> RefCritic adopts a dual-reward mechanism. We first require the Critic to make an accurate binary judgment on solution correctness, which is a prerequisite for effective feedback (if the judgment is wrong, the reward is set to 0). We calculate the Refinement Reward only if the initial response is incorrect and the Critic correctly identifies the error. This design ensures the critic possesses both accurate judgment and effective guidance capability.
>
> ---
>
> **3. Critique Generation Form (Standard Text Feedback vs. Long Chain-of-Thought Critic)**\
> RCO generates standard critiques where all critique is returned to the Policy model. RefCritic trains a Long Chain-of-Thought Critic. As shown in the template in Appendix G, we separate the Critic's reasoning process from the feedback provided to the Policy Model.
> Through reinforcement learning, RefCritic-R1's average output length increased from 3,000 to 8,000 tokens, with the growth primarily occurring in the reasoning process rather than the feedback section. This offers two key benefits: 1) The Deep Critique form enables RefCritic to achieve step-level error localization without step-level supervision (as shown in the ProcessBench experiments in Section 5.4). 2) The separated feedback part allows us to provide only a concise portion of the critique to the Policy Model, preventing the Policy Model's Refinement from being misled by the Critic's internal bias during evaluating the wrong solution. This design enables RefCritic to effectively guide stronger Policy Models (as shown in Section 5.6: SUPERVISION OF STRONGER MODELS) and iterative refinement (as shown in Section 5.5: TEST-TIME SCALING).
>
> ---
>
> In summary, while both RefCritic and RCO utilize refinement signals, there are substantial differences between them in multiple aspects. We will incorporate RCO into the related work discussion in the revised version. We thank the reviewer for helping us clarify RefCritic's unique contributions.

---

> ### Author Response · Authors · 2025-11-22
> **Response to Reviewer qP4v (Part 2/2)**
>
> > **W2. Although the authors claim that their method is a long-chain-of-thought critic module, I do not find how this method can improve the long-chain-of-thought generation ability...**
> ---
>
> Although we did not explicitly propose a specific architectural design for long-chain-of-thought in the methodology section, RefCritic's long-chain-of-thought capability is demonstrated in multiple aspects.
>
> First, as stated in Section 5.4 CRITIC PERFORMANCE, the average output length of RefCritic-Qwen increased from about 500 tokens to 3,500 tokens, while RefCritic-R1 increased from 3,000 tokens to 8,000 tokens. This indicates that RefCritic naturally exhibits long-chain-of-thought capabilities after two-stage training. In fact, as demonstrated by recent influential RL works such as [1] and [2], long chain-of-thought reasoning can naturally emerge through RLVR.
>
> Furthermore, during Refinement, we do not return the full Critique to the Policy Model. As shown in the templates in Appendix G, we separate the Critic's reasoning process from the Feedback provided to the Policy Model. By internalizing the format via SFT and employing two-stage RL, this design enables RefCritic to naturally emerge with long Critique analysis alongside concise and effective Feedback (as shown in the Case Study in Appendix F). This behavior significantly promotes RefCritic's performance when providing feedback to other models, especially larger-scale ones. A longer critique analysis helps more precisely identify errors in the solution and generates more effective feedback, while concise feedback prevents the Policy Model from being influenced by biases in the Critic's reasoning. This partly explains RefCritic's excellent performance in Section 5.4 CRITIC PERFORMANCE and Section 5.6 SUPERVISION OF STRONGER MODELS.
>
> [1] DeepSeek-R1: Incentivizing Reasoning Capability in LLMs via Reinforcement Learning \
> [2] SimpleRL-Zoo: Investigating and Taming Zero Reinforcement Learning for Open Base Models in the Wild
>
> ---
> >**W3. The quality of generated critiques themselves should be assessed via automatic metrics or human evaluation.**
> ---
>
> Thank you for your suggestion. We assess the quality of generated critiques using both automatic metrics and human evaluation:
>
> * **Automatic Metrics:** We measure performance from both coarse-grained and fine-grained accuracy perspectives.
>     * **Coarse-grained accuracy:** We investigated RefCritic's judgment accuracy on Solutions in Section 5.5. As shown in Figure 2(a), RefCritic-R1-14B achieves an accuracy of over 86% on AIME25 for solutions from DeepSeek-R1-Distill-Qwen-14B, and reaches up to 88.5% on Olympiad (surpassing 90% after scaling the number of critiques). This indicates that RefCritic achieves very high performance in coarse-grained judgment.
>     * **Fine-grained accuracy:** We presented RefCritic's performance on ProcessBench in Section 5.4 CRITIC PERFORMANCE. ProcessBench requires the Critics to identify the specific step where the first error occurs. This fine-grained error step localization is effectively Out-of-Distribution for RefCritic, as we do not use step-level rewards during training. Nevertheless, RefCritic still outperforms other baselines, demonstrating strong fine-grained accuracy.
>
> * **Human Evaluation:** We added the case study section in Appendix F, illustrating the difference between Critics trained with RefCritic and those trained using GRPO. Critics trained with RefCritic can accurately localize errors and offer guidance from a high-level perspective, providing more reasonable and correct instructional feedback.
>
> ---
> > **W4. The content of Line 262 should be after that of Line 249. The explanation of \lambda should be after the appearance.**
> ---
>
> Thank you for pointing this out. We will revise this section and move the relevant explanation to the appropriate position. You can find the refined explanation in Section 4 REFCRITIC of the newly revised PDF.

---

### Official Review · Reviewer_qX2t · 2025-11-09

**Soundness:** 2
**Presentation:** 3
**Contribution:** 2
**Rating:** 4
**Confidence:** 4

**Summary:**

This paper proposes RefCritic, a training framework for long chain of thought critics that combines supervised fine-tuning with rule-based reinforcement learning. The critic is optimized with two rewards, one for accurate solution-level correctness judgment and the other for the accuracy gain when the policy solves the problem based on the critique. The goal is to make critiques not only label answers as right or wrong but also provide feedback that actually improves the policy. Experiments on Qwen2.5-14B-Instruct and DeepSeek-R1-Distill-Qwen-14B show consistent improvements on AIME24, AIME25 and Olympiad style benchmarks, for example, an absolute improvement of around 7 percentage points on AIME25 in terms of Passr@1. Although training only uses solution-level labels, RefCritic also outperforms several step-supervised baselines on step level error detection in ProcessBench.

**Strengths:**

1. The paper is logically clear and easy to follow.

2. The use of refinement effectiveness as a direct reward for the critic is conceptually clear and technically reasonable.

3. Experiments are extensive, including test time scaling, comparisons with multiple base models and several out-of-distribution benchmarks.

**Weaknesses:**

1. The introduction and main experiments lack systematic comparison with recent strong critic baselines such as DeepCritic [1] and RealCritic [2], so the position and advantage of RefCritic in this line of work remain unclear.

2. The paper does not report the computational cost of the dual reward RL training, and hence it is difficult to assess the cost effectiveness and scalability of the proposed approach to larger models or broader deployment.

3. The paper lacks qualitative cases that show critic outputs and policy answers before and after refinement, which makes it harder to see how RefCritic behaves in practice.

4. The ablation study on design choices seems limited. Beyond lambda, other important hyperparameters such as the number of critiques and refinements per example, the two-stage RL schedule, and the decoding and length settings are not analyzed, so it is unclear the sensitivity of the proposed method is to these choices.

[1] Yang W, Chen J, Lin Y, et al. DeepCritic: Deliberate Critique with Large Language Models[EB/OL]. arXiv:2505.00662, 2025.

[2] Tang Z, Li Z, Xiao Z, et al. RealCritic: Towards Effectiveness-Driven Evaluation of Language Model Critiques[EB/OL]. arXiv:2501.14492, 2025.

**Questions:**

1. The rewards Rc and λRr rely on a rule-based discriminator that checks whether the generated answer matches the golden answer. Could the authors clarify whether this discriminator is robust to mathematically equivalent but differently formatted expressions, and please also provide the sensitivity analysis for the results of this choice?

2 .On page 3, the footnote for Qwen2.5 14B Instruct says “we provide an empty thinking process and only use the content after ”¡/think¿””. This tag and quoting style look unusual compared to the usual </think> format. Could the authors clarify what exact tag and quotation you use here?

3. In Figure 1, the RL training reward includes a factor like (1 + 0.75) / (1 + λ). Could the authors explain the motivation for this normalization and in particular why the reward is divided by 1 + λ?

4. In Appendix B, some ablation results appear different from the corresponding main results. Could the authors clarify the differences in experimental setup and explain this discrepancy in detail?

---

> ### Author Response · Authors · 2025-11-22
> **Response to Reviewer qX2t (Part 1/4)**
>
> > **W1. The introduction and main experiments lack systematic comparison with recent strong critic baselines such as DeepCritic [1] and RealCritic [2], so the position and advantage of RefCritic in this line of work remain unclear.**
> ---
>
> Thank you for your suggestion. To better contextualize RefCritic among related works, we have incorporated DeepCritic and ThinkPRM—two influential baselines—into our main experiments. Their performance as standalone critics was already reported in Section 5.4. We exclude SCRIT-72B from comparison as its weights are not open-sourced. The detailed results are as follows:
>
> Qwen-14B as policy model
> | Model | AIME25 $Pass_r@1$ | AIME25 $Maj_c@8$ | AIME25 $Maj_c@64$ | AIME24 $Pass_r@1$ | AIME24 $Maj_c@8$ | AIME24 $Maj_c@64$ | Olympiad $Pass_r@1$ | Olympiad $Maj_c@8$ | Olympiad $Maj_c@16$ |
> | :--- | :---: | :---: | :---: | :---: | :---: | :---: | :---: | :---: | :---: |
> | Qwen-14B Majority Vote *(No Critic)* | 14.4 | 19.2 | 23.3 | 13.7 | 16.5 | 16.6 | 45.8 | 52.2 | 53.6 |
> | Qwen-14B Self-Critic | 14.5 | 19.1 | 22.7 | 13.7 | 18.5 | 21.2 | 45.8 | 52.3 | 54.0 |
> | DeepCritic-7B | 13.0 | 17.4 | 19.4 | 13.4 | 18.8 | 23.7 | 43.6 | 51.7 | 52.7 |
> | ThinkPRM-14B | 16.1 | 20.1 | 22.3 | 16.8 | 20.7 | **26.9** | 49.7 | 54.7 | 56.6 |
> | RefCritic-Qwen-14B(SFT) | 15.0 | 19.3 | 23.4 | 15.2 | 19.2 | 23.9 | 46.6 | 52.5 | 54.3 |
> | RefCritic-Qwen-14B(RL) | **21.2** | **21.5** | **24.4** | **23.0** | **21.4** | 26.6 | **55.7** | **57.3** | **59.2** |
>
> R1-Qwen-14B as policy model
> | Model | AIME25 $Pass_r@1$ | AIME25 $Maj_c@8$ | AIME25 $Maj_c@64$ | AIME24 $Pass_r@1$ | AIME24 $Maj_c@8$ | AIME24 $Maj_c@64$ | Olympiad $Pass_r@1$ | Olympiad $Maj_c@8$ | Olympiad $Maj_c@16$ |
> | :--- | :---: | :---: | :---: | :---: | :---: | :---: | :---: | :---: | :---: |
> | R1-Qwen-14B Majority Vote *(No Critic)* | 49.1 | 61.6 | 62.0 | 67.6 | 78.7 | 80.1 | 77.7 | 82.7 | 83.3 |
> | R1-Qwen-14B Self-Critic | 50.0 | 60.6 | 62.9 | 70.5 | 79.3 | 82.4 | 78.8 | 82.7 | 83.3 |
> | DeepCritic | 49.1 | 58.0 | 59.0 | 67.3 | 77.2 | 78.8 | 76.1 | 81.6 | 82.1 |
> | ThinkPRM-14B | 43.9 | 58.7 | 61.2 | 62.6 | 76.3 | 81.0 | 75.2 | 82.1 | 82.7 |
> | RefCritic-R1-14B(SFT) | 51.3 | 61.6 | 62.8 | 71.4 | 79.4 | **83.1** | 78.7 | 83.0 | 84.4 |
> | RefCritic-R1-14B(RL) | **56.3** | **65.2** | **68.1** | **72.8** | **80.4** | 82.5 | **80.3** | **83.9** | **84.7** |
>
> Our experiments demonstrate that RefCritic not only outperforms all baselines in critique tasks but also achieves superior performance when interacting with the Policy Model. We will include these results in Section 5.2 MAIN RESULTS in the revised PDF.

---

> ### Author Response · Authors · 2025-11-22
> **Response to Reviewer qX2t (Part 2/4)**
>
> > **W2. The paper does not report the computational cost of the dual reward RL training, and hence it is difficult to assess the cost effectiveness and scalability of the proposed approach to larger models or broader deployment.**
> ---
>
> We agree that computational cost is an important topic for discussion. Our analysis indicates that while RefCritic incurs approximately $1\times$ additional computational cost compared to training without Refinement Reward, it delivers significant performance improvements—averaging about 10% gains across three mathematical benchmarks (detailed in Appendix B ABLATION). This validates the value of the additional computational investment. The detailed cost estimation is as follows:
>
> Please note that we employ two strategies to control training costs:
> 1. RefCritic only performs Refinement on solutions that are both incorrect and judged as incorrect by the Critic.
> 2. Refinement Reward is not used during the initial phase of reinforcement learning.
>
> **Symbol Definitions:**
>
> | Symbol | Definition |
> |------|------|
> | $c_c$ | Cost of a single Critic rollout during GRPO training |
> | $c_r$ | Cost of a single Refinement rollout during GRPO training |
> | $r_c$ | Number of Critique rollouts per solution |
> | $r_r$ | Number of Refinement rollouts per solution |
> | $a$ | RefCritic's judgment accuracy on incorrect solutions |
> | $m$ | Number of training steps without Refinement Reward |
> | $n$ | Number of training steps with Refinement Reward |
>
> Assuming both standard GRPO and RefCritic are trained for $m+n$ steps:
> * **Cost of standard GRPO:** $Cost_{GRPO} = (m+n)r_cc_c$
> * **Cost of RefCritic (Phase 1, $m$ steps, no Refinement Reward):** $mr_cc_c$
> * **Cost of RefCritic (Phase 2, $n$ steps, with Refinement Reward):** $nr_cc_c+(an/2)r_cr_rc_r$
>     * $nr_cc_c$ represents the cost of critique rollouts, and the second term represents the additional refinement cost. Since we maintain a balanced RL dataset with equal proportions of correct and incorrect solutions, there are $an/2$ samples requiring refinement rollouts. Thus, the actual refinement cost is $anr_cr_rc_r/2$.
>
> The ratio $T$ of additional refinement cost to original cost (only critique cost) is:
> $$T=\frac{anr_cr_rc_r}{2(m+n)r_cc_c}=\frac{anr_rc_r}{2(m+n)c_c}=a\frac{n}{2m+2n}r_r\frac{c_r}{c_c}$$
>
> In our experiments, we set $m:n=2:1$ and $r_r=8$. Based on empirical observation, $a$ can be approximated as 0.8. Thus:
> $$T=0.8 \times \frac{1}{6} \times 8 \times \frac{c_r}{c_c} = \frac{16}{15} \times \frac{c_r}{c_c}$$
>
> Regarding $\frac{c_r}{c_c}$, we can reasonably assume it is close to 1. As noted in Section 5.4 CRITIC PERFORMANCE, the average output length of RefCritic-Qwen increased from approximately 500 to 3500 tokens, while RefCritic-R1 increased from 3000 to 8000 tokens. Besides, we do not provide the full Critique output to the Policy Model for Refinement (only the feedback part, you can find the exact template in Appendix F and G). Given that the Critic's generation length is substantially longer than the Policy Model's Refinement, the actual value of this ratio should be lower.
>
> Thus, we can conclude:
> $$T \approx 1$$
>
> This indicates that RefCritic with Refinement Reward brings only about $1\times$ additional computational cost. We will include this detailed cost analysis in **Appendix E** of the revised PDF.

---

> ### Author Response · Authors · 2025-11-22
> **Response to Reviewer qX2t (Part 3/4)**
>
> > **W3. The paper lacks qualitative cases that show critical outputs and policy answers before and after refinement, which makes it harder to see how RefCritic behaves in practice.**
> ---
>
> Thank you for this suggestion. We have added case studies in **Appendix F** that demonstrate the differences between critics trained with Refinement Reward and those trained with standard GRPO (only critique reward).
>
> Critics trained with Refinement Reward are more capable of accurately localizing errors and providing reasonable, correct guidance. In contrast, critics without Refinement Reward training, although potentially achieving correct binary judgments, often struggle to offer constructive suggestions and tend to provide unsuitable advice influenced by the erroneous solution.
>
> ---
> > **W4. The ablation study on design choices seems limited. Beyond lambda, other important hyperparameters...**
> ---
>
> Regarding your concerns about the ablation study, we need to emphasize that RefCritic's primary research objective is to establish a robust two-stage reinforcement learning pipeline that trains a critic capable of both accurately judging solution correctness and providing valuable feedback.
>
> Our main goal is to validate the effectiveness of the **Refinement Reward**. Other hyperparameters are not the central focus of this investigation. For the number of critiques and refinements: intuitively, using more during RL should yield higher returns. However, such improvements would be attributable to increased rollouts in RLVR algorithms like GRPO, as extensively studied in prior work [1], rather than to RefCritic's specific design. Furthermore, simply conducting ablations on decoding strategies and length settings would not provide deeper insights into our contributions; even identifying optimal settings would not validate any conclusions specific to this paper. Therefore, conducting extensive RL exploration on these general parameters would yield limited returns.
>
> Finally, we have already conducted comprehensive ablation studies on the core component of RefCritic—the **Refinement Reward** and its corresponding parameter $\lambda$. Other training hyperparameters are beyond the scope of this research.
>
> [1] Hu, J., Liu, M. Brorl: Scaling reinforcement learning via broadened exploration. arXiv preprint arXiv:2510.01180, 2025b.

---

> ### Author Response · Authors · 2025-11-22
> **Response to Reviewer qX2t (Part 4/4)**
>
> > **Q1. The rewards Rc and λRr rely on a rule-based discriminator that checks whether the generated answer matches the golden answer...**
> ---
>
> We understand your concern about the reward signal reliability. We utilize the same rule-based discriminator as the official **Qwen-Math** implementation ([https://github.com/QwenLM/Qwen2.5-Math](https://github.com/QwenLM/Qwen2.5-Math)). Similar discriminators have been widely accepted and utilized in Math-RL research [1, 2, 3].
>
> This discriminator is built on sophisticated code logic and exhibits high robustness. It leverages Python libraries such as SymPy and latex2sympy to convert expressions into **canonical forms** for comparison, rather than performing simple string matching. Consequently, it effectively handles mathematically equivalent but differently formatted expressions, following pairs of expressions are considered equivalent:
> * $g(x)=x^2-3x+2$ = $(x-2)(x-1)$
> * $\frac{17}{4}$ = $4.25$
> * $18.8$ = $18.8^{\circ}$
>
> Furthermore, our rigorous data filtering pipeline (detailed in Appendix C DATA CONSTRUCTION) further minimizes the probability of misjudgment, ensuring that for most problems, the answers are parsable by the discriminator and generatable by the LLM. These settings ensure we obtain robust rewards during critic training.
>
> [1] DAPO: An Open-Source LLM Reinforcement Learning System at Scale \
> [2] Qwen2.5-Math Technical Report: Toward Mathematical Expert Model via Self-Improvement \
> [3] Does Reinforcement Learning Really Incentivize Reasoning Capacity in LLMs Beyond the Base Model?
>
> ---
> > **Q2. On page 3, the footnote for Qwen2.5 14B Instruct...**
> ---
>
> Thank you for your careful observation. There is no special formatting intended here; the issue arose because Overleaf's special syntax incorrectly escaped "`<think>`". We have corrected this issue in the revised PDF.
>
> ---
> > **Q3. In Figure 1, the RL training reward includes a factor like (1 + 0.75) / (1 + λ). Could the authors explain the motivation for this normalization and in particular why the reward is divided by 1 + λ?**
> ---
>
> This design follows a weighted average philosophy, ensuring that the maximum possible total reward remains 1 when considering both the Critique Reward and the Refinement Reward.
>
> As explained in the paper, we set $\lambda$ as the weight of Refinement Reward relative to Critique Reward. Since RefCritic only calculates Refinement Reward for solutions that are **incorrect and correctly identified as incorrect by the Critic**, the denominator in the figure is $1+0.75\lambda$ (where 0.75 represents the Refinement accuracy).
>
> To prevent reward imbalance during reinforcement learning, we divide the reward in this scenario by $1 + \lambda$. Without this normalization, critiques of incorrect solutions could receive higher rewards than those of correct solutions, leading the model to develop a bias toward judging solutions as incorrect after training. Under this design, the model achieves the maximum reward if and only if it correctly judges the correctness of all solutions and provides the highest-quality critiques for all incorrect solutions.
>
> ---
> > **Q4. In Appendix B, some ablation results appear different from the corresponding main results. Could the authors clarify the differences in experimental setup and explain this discrepancy in detail?**
> ---
>
> We have not identified the discrepancies you mentioned. We would be very grateful if you could specify the issues so we can address them appropriately.

---

### Author Response · Authors · 2025-12-02
**Rebuttal Summary**

We would like to thank the Area Chairs for their hard work during this exceptional period. **We respectfully hope that Area Chair could consider the substantial improvements and experimental evidence made during the rebuttal period, the positive trajectory of our discussions with the reviewers, when making the final decision.**

We sincerely thank all reviewers for their constructive feedback and positive engagement. We have addressed every concern with detailed explanations and comprehensive supplementary experiments. All supplementary contents have been incorporated into the revised PDF.

**We are pleased to report significant progress during rebuttal:**
- **Reviewer XRcp (4->6)**  confirming our responses "largely addressed their concerns," and raised their score to 6.
- **Reviewer wJJH (2->4)** acknowledged our detailed response, addressed their concerns, and raised the score to 4.
- **Reviewers qX2t (4) and qP4v (6)** have not had the chance to respond, but we provided substantial additional experiments as requested. We are confident that these comprehensive responses can lead to further score increases if the rebuttal process continues as initially planned.

**We would like to emphasize that the reviewer's responses and score raising all occurred before the data leak.**

To facilitate the AC's review, we summarize reviewers' key concerns and our responses below:

---

## **Reviewer qX2t (Score: 4 → Waiting for response)**
---

- **W1: Request for comparison with strong baselines**
- R: We integrated **DeepCritic** and **ThinkPRM** into the main experiments (Table 2). Results demonstrate that RefCritic consistently outperforms all baselines across all settings.
---
- **W2: Missing computational cost analysis**
- R: We provided a detailed analysis in Appendix E. Results prove RefCritic incurs only **~1$\times$** additional cost compared to standard GRPO while delivering **~10%** average performance improvement.
---
- **W3: Need for qualitative case studies**
- R: We added comprehensive case studies in Appendix F, demonstrating RefCritic accurately localizes errors and provides actionable feedback.
---
- **W4: Limited ablation study on hyperparameters**
- R: We prioritized ablating the core **Refinement Reward**. Extensive ablations on general RL parameters offer limited value specific to our contribution.
---
- **Q1: Robustness of rule-based discriminator**
- R: We use the same discriminator as the official Qwen-Math implementation, which is widely accepted in Math-RL research.
---
- **Q3: Motivation for reward normalization factor (1+0.75λ)/(1+λ)**
- R: We explained that this normalization ensures the maximum total reward remains 1, preventing reward hack.

---

## **Reviewer qP4v (Score: 6 → Maintaining positive)**
---

- **W1: Comparison with RCO (Training LLM to Critique for Better Refinement)**
- R: Clarified three key differences from RCO: (1) **Dual-objective** vs. single-objective, (2) **Objective ground truth** vs. preference-based reward, (3) **Long CoT with separated reasoning and feedback** enabling step-level localization without step-level supervision.
---
- **W2: Long Chain-of-Thought capabilities**
- R: Demonstrated through two-stage RL, output length naturally scales (**500→3,500 tokens for Qwen**, **3,000→8,000 tokens for R1**), with growth primarily in the critique analysis process, enabling fine-grained error detection.
---
- **W3: Critique quality assessment**
- R: We validated quality via both automatic metrics and human evaluation (Appendix F).
---

## **Reviewer XRcp (Score: 4 → 6)**
---

- **W1: Comparison with Multi-agent Feedback (MultiCritique)**
- R: We clarified that RefCritic focuses on **Deep CoT** rather than multi-agent aggregation. Our method uses refinement directly in the RL training phase.
---
- **W2: Reliance on verifiable ground truth**
- R: Clarified RLVR(Reinforcement Learning with Verifiable Rewards) is gaining significance for general domains and shows transferable cognitive abilities for open-ended tasks.
---
- **W3: Computational cost**
- R: As qX2t's W2.

---

## **Reviewer wJJH (Score: 2 → 4)**
---

- **W1: Comparison with CTRL (Teaching Language Models to Critique via Reinforcement Learning)**
- R: Clarified three major differences from CTRL: (1) **Dual-objective optimization**, (2) **Judgment-first design** enabling weighted majority voting and avoiding over-refinement of correct solutions, (3) **Novel insights**: step-level localization and strong OOD transfer.
---
- **W2(1)/Q1: Missing Baselines**
- R: As qX2t's W1
---
- **W2(2): Deviation from ProcessBench evaluation protocol**
- R: Clarified extractor (Qwen-14B) has **96% accuracy** (manual evaluation of 50 cases) and does not overestimate performance.
---
- **Q2: Multi-turn iterative refinement exploration**
- R: Added extensive experiments in Section 5.5 showing consistent improvement across 8 rounds: **AIME24: 13.6%→27.2%**, **Olympiad: 45.7%→60.1%**

---

### Meta-Review · Area_Chair_i298 · 2026-01-08

**Summary:**

This paper proposes RefCritic, a method for training critic models using reinforcement learning with dual rewards: (1) instance-level solution correctness judgments and (2) refinement accuracy improvements when policy models use the critic's feedback.

The reviewers raised several concerns: (1) concerns about novelty given conceptual similarities to existing work; (2) missing comparisons with strong recent baselines like DeepCritic, ThinkPRM, and related work such as RCO, CTRL, and MultiCritique; (3) insufficient computational cost analysis for the dual-reward RL training; (4) lack of qualitative case studies demonstrating critic behavior; and (5) methodological deviations in ProcessBench evaluation.

The reviewers found the paper clearly written but raised significant concerns regarding novelty and experimental methodology that doesn't seem fully resolved during the rebuttal period.

**Reviewer Concerns:**

Addressed Concerns: baseline comparisons, computational cost, case studies, related work discussion. Outstanding Concerns: novelty issue, ProcessBench evaluation deviations.

**Reviewer Scores:**

Reviewer qX2t: 50% = 4, 50% -> 6
Reviewer qP4v: = 6
Reviewer XRcp: 4 -> 6
Reviewer wJJH: 2 -> 4

---

### Decision · Program_Chairs · 2026-01-26

Reject